# Nonlinear variations and drivers of vegetation NPP on the Tibetan Plateau: Interaction of natural and human factors

Jie Tang[1,2], Xinghong Peng[3], Wenfu Peng[1,2]*

1 The Institute of Geography and Resources Science, Sichuan Normal University, Chengdu, China, 2 Key Lab of Land Resources Evaluation and Monitoring in Southwest, Ministry of Education, Chengdu, China, 3 Sichuan Land Consolidation and Rehabilitation Center, Chengdu, China

☉ WP contributed equally to this work.
* pwfzh@126.com

## Abstract

Understanding the drivers of changes in vegetation net primary productivity (NPP) is critical for comprehending ecosystem dynamics and their ability to respond to environmental shifts. However, the complexity and nonlinear variations of NPP across the Tibetan Plateau, along with spatial and temporal inconsistencies, present significant analytical challenges. This study leverages the Google Earth Engine (GEE) platform and applies non-parametric trend analysis methods, such as the Sen slope estimator, Mann-Kendall test, coefficient of variation, and Hurst exponent, to investigate NPP trends from 2001 to 2021. The Optimal Parameters-based Geographical Detector (OPGD) model was employed to assess the combined effects of natural factors and human activities on NPP's spatial distribution and variability, identifying key drivers and their optimal ranges for promoting NPP growth. Results revealed nonlinear fluctuations in NPP during the study period, ranging from 184.06 to 208.53 gC m$^{-2}$.a$^{-1}$, with an average annual growth rate of 1.16 gC m$^{-2}$.a$^{-1}$. Significant spatial differences were observed, with higher NPP in the grasslands and forests of the southeast, while lower productivity was found in the alpine deserts of the northwest. Over 55% of the study area showed an increasing trend in NPP, with 28.14% experiencing significant growth ($p < 0.05$). The study further indicated that natural factors such as elevation, solar radiation, and mean annual temperature were major determinants of NPP fluctuations, while human activities (e.g., distance, population density, and land use) also played a crucial role in shaping NPP patterns. The significant interaction between natural factors and human activities demonstrates synergistic enhancement and non-linear effects, highlighting the complexity of multi-factor drivers influencing NPP changes. The key promoting factors and their optimal ranges identified provide a foundation for understanding the impact of natural and human activities on NPP variation, offering scientific support for ecosystem management and sustainable development on the Tibetan Plateau.

**Data availability statement:** We confirm that all datasets are publicly available for research use and verify their accuracy. The relevant datasets can be accessed through the provided links: 1. Vegetation NPP, soil moisture, evapotranspiration, and total solar radiation data are sourced from the Earth Engine data catalogue (https://developers.google.com/earth-engine/datasets). 2. China's administrative boundary data (1:4 million scale), natural factor data (including DEM, temperature, accumulated temperature ≥0°C, and precipitation), and human activity data (such as land use types, GDP density, and population density) are obtained from the Data Centre for Resources and Environmental Sciences, Chinese Academy of Sciences (RESDC) (http://www.resdc.cn). 3. Qinghai-Tibet Plateau boundary data are from the study "On the Extent and Area of the Qinghai-Tibet Plateau: Geographic Information System Data on the Boundaries and Area of the Qinghai-Tibet Plateau" published by Zhang et al. (2014) (http://www.geodoi.ac.cn/doi.aspx?doi=10.3974/geodb.2014.01.12.v1).

**Funding:** This work was supported by the National Ministry of Education Humanities and Social Sciences Research Planning Fund Project, China (Grant No. 17YJA850007), titled Regional Differentiation of Poverty and Strategies for Targeted Alleviation in Tibetan Areas of the Northwest Sichuan Plateau. The funders had no role in study design, data collection, analysis, decision to publish, or manuscript preparation.

**Competing interests:** This manuscript has neither been published elsewhere nor is it under consideration for publication elsewhere, in whole or in part. All authors have made substantial contributions to the study, reviewed and approved the manuscript, and agreed to its submission to the journal. The authors declare no competing interests.

## Introduction

Vegetation net primary productivity (NPP) is a crucial indicator of an ecosystem's carbon sequestration capacity and overall productivity [1]. It serves as a vital metric for understanding ecosystem health and sustainability, particularly in the context of global environmental changes. NPP variations not only mirror environmental changes but also reflect the impact of human activities on ecosystems [2,3,4]. The Tibetan Plateau, as a sensitive region under global climate change, provides an ideal setting to study the intricate dynamics of NPP and its drivers [5,6].

Recent studies have highlighted significant spatiotemporal variations in NPP across the Tibetan Plateau. Vegetation NPP has generally increased since 2000, particularly in the humid eastern regions, while changes in the arid western areas remain minimal [7]. These variations are strongly linked to regional climatic conditions, emphasizing the need to better understand these patterns to predict future ecological changes under global climate scenarios [8].

Climatic factors, especially temperature and precipitation, are identified as the primary drivers of NPP dynamics, with their effects varying significantly across subregions [9,10]. Recent findings further reveal that precipitation dominates the interannual variability of vegetation productivity, particularly in arid regions, while temperature exerts a stronger influence in alpine and humid areas [8,11]. Additionally, solar radiation has been found to interact with other climatic variables, influencing vegetation growth patterns on the plateau [12,13].These findings collectively highlight the intricate interplay between climatic drivers and vegetation productivity across the Tibetan Plateau, offering critical insights for forecasting and managing ecosystem responses to ongoing climate change.

Despite significant advancements, existing research still exhibits notable gaps. First, while many studies emphasize linear relationships between NPP and its drivers, the potential interactive and nonlinear effects remain insufficiently explored. For instance, the combined impacts of precipitation, temperature, and solar radiation on vegetation productivity are often simplified, despite their known complex interactions [8,12]. Limited research has examined the synergies and trade-offs between climatic and anthropogenic factors, as these are frequently treated as independent drivers [10].

Second, the spatiotemporal heterogeneity of NPP remains underexplored. Although several studies have addressed localized regions or short-term trends, a lack of systematic, long-term investigations across the entire plateau has hindered a comprehensive understanding of these variations [13]. Recent studies highlight the importance of addressing this gap, particularly through high-resolution datasets and advanced modeling techniques [14].

Third, while human activities such as overgrazing, urban expansion, and land-use changes are acknowledged in some studies, there is a critical need for systematic analyses that integrate these factors with natural drivers across varying spatial and temporal scales [11]. Such integrated approaches are crucial for quantifying the combined impacts of natural and human-induced changes on NPP dynamics.

Lastly, the Tibetan Plateau's complex terrain and diverse ecosystems pose significant challenges for comprehensive analyses. Advances in remote sensing and machine learning techniques offer promising avenues to overcome these limitations, enabling more accurate and detailed assessments of NPP dynamics across diverse ecological zones [15].

Google Earth Engine (GEE), a powerful cloud-based platform, integrates a wide range of open-source geospatial datasets and offers an accessible programming interface and API [16]. This study aims to address these gaps by leveraging Google Earth Engine (GEE) and the OPGD model. GEE's capacity for integrating multi-source remote sensing data with high computational efficiency makes it particularly suitable for large-scale studies in remote and ecologically fragile regions like the Tibetan Plateau. GEE offers efficient processing of large-scale geospatial datasets, while the OPGD model quantifies the interactive effects of natural and anthropogenic factors, enabling precise spatiotemporal analysis [17–19].

The objectives of this study are as follows: (1) process NPP imagery for the Tibetan Plateau using GEE and evaluate monotonic trends with the Sen slope and Mann-Kendall test, (2) analyze the interactions between natural and human factors influencing NPP variations using the OPGD model to enhance spatial analysis accuracy, (3) identify optimal conditions for NPP growth to support ecological management on the Tibetan Plateau.

## Study area, data sources and research methods

### Study area

The Tibetan Plateau, located within China, stretches from the Pamir Plateau in the west to the Hengduan Mountains in the east, from the southern edge of the Himalayas in the south to the northern side of the Kunlun-Qilian Mountains in the north. Its geographical range spans from 25°59′37″N to 39°49′33″N and 73°29′56″E to 104°40′20″E, covering an area of 2.5423 million km² [20]. The Tibetan Plateau is the highest plateau in the world, with an average altitude of approximately 4,000 metres. The region features diverse natural environments, with annual mean temperatures ranging from -6 °C to 20 °C and precipitation varying between 50 mm and 2000 mm. These unique geographical and climatic conditions have shaped a wide variety of ecosystems, with vegetation types transitioning from subtropical rainforests in the southeast to alpine deserts in the northwest, exhibiting a distinct latitudinal and vertical zonation [21,22].

In China's vegetation classification system, the Tibetan Plateau is divided into 11 vegetation zones, with alpine grasslands (including both steppes and meadows) covering 60% of the plateau [23]. The Tibetan Plateau is not only a vital component of global ecological balance and the climate system, but it is also a sensitive indicator of climate change, playing a crucial role in the water supply to South and Central Asia. In recent years, the average temperature of the Tibetan Plateau has risen by 1.9 °C, a warming rate double the global average, and precipitation levels have also increased (Fig 1).

### Data sources and processing

The variations in NPP across the Tibetan Plateau can be scientifically elucidated by integrating both natural factors and human activities [24,25]. Natural elements such as topography, climate, and moisture conditions form the foundational basis for changes in NPP. Meanwhile, human activities, particularly land use and development intensity, further modify the spatial distribution and dynamic patterns of NPP [20,26–28]. Consequently, this study focuses on selecting these critical natural and human factors to accurately assess the spatial variation of NPP influenced by both environmental conditions and human interventions across the Tibetan Plateau.

This research utilizes a variety of datasets, including vegetation NPP, natural factors, and human activity indicators pertinent to the Tibetan Plateau (refer to Table 1). Vegetation NPP data were obtained from the Earth Engine Data Catalog (MODIS/061/MOD17A3HGF) and processed through Google Earth Engine (GEE) scripts, providing a spatial resolution of 500 meters. The natural factors considered include digital elevation model (DEM), slope, aspect, soil moisture, solar radiation, annual mean temperature, accumulated temperature above 0°C, annual precipitation, and evapotranspiration.

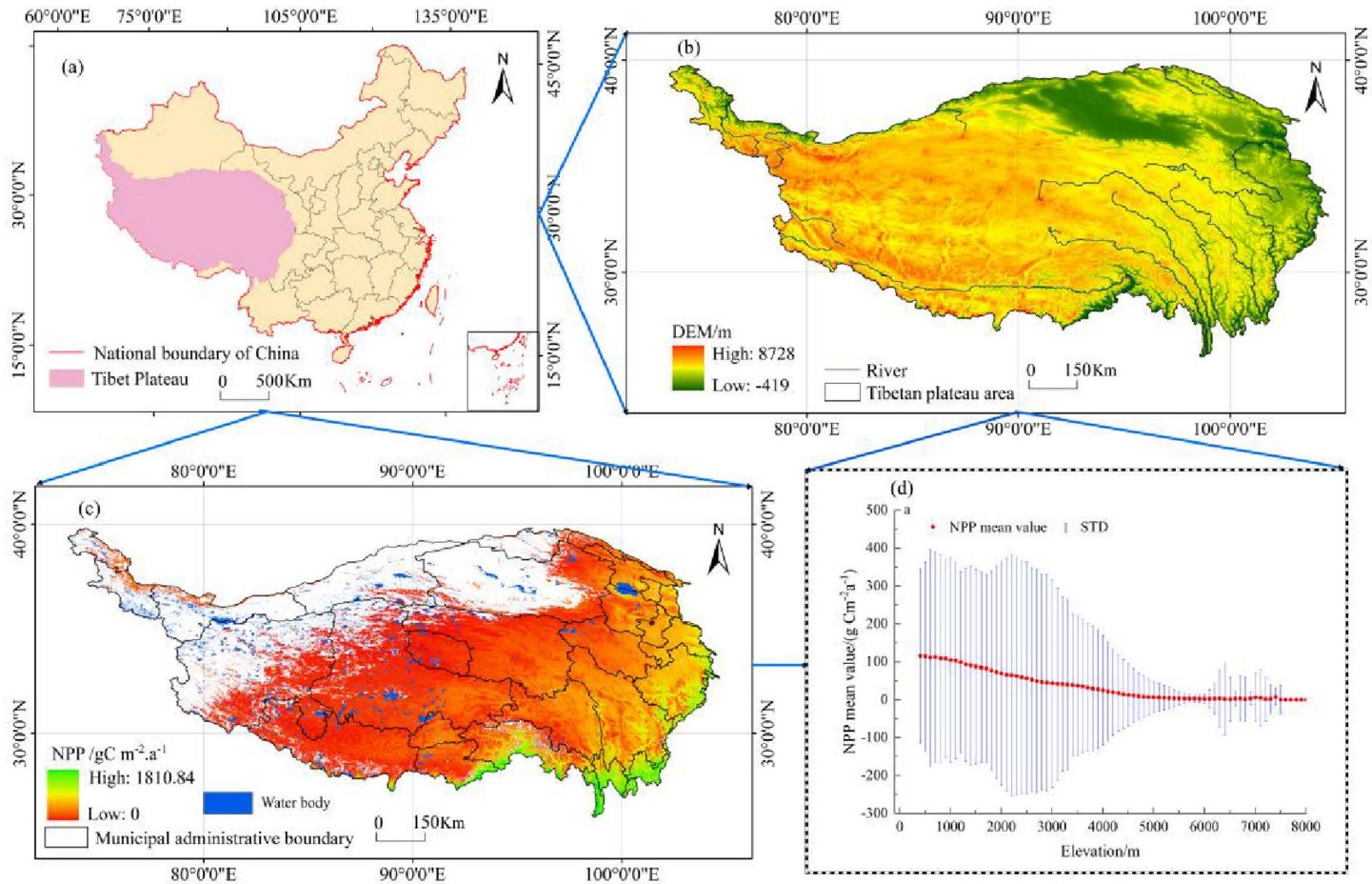

**Fig 1. Location of the study area** : (A) The location of the Tibetan Plateau within China; (B) Digital Elevation Model (DEM); (c) mean vegetation NPP on the Tibetan Plateau from 2001 to 2021; (D) variation of NPP with elevation from 2001 to 2021.

DEM, temperature, and precipitation data were sourced from the Data Centre for Resources and Environmental Sciences, Chinese Academy of Sciences (https://www.resdc.cn/), while slope and aspect were derived from DEM using GIS tools, maintaining a spatial resolution of 100 meters. The remaining natural factors were also retrieved via GEE scripts, with a resolution of 500 meters.

The dataset on human activities comprised variables such as land use types, GDP density, and population density. Most human activity datasets, excluding those related to rivers and transportation, were also sourced from the Data Centre for Resources and Environmental Sciences, Chinese Academy of Sciences (https://www.resdc.cn/). Variables regarding land use intensity, proximity to cropland, forest, grassland, water bodies, road density, and road distance were extracted using GIS tools, with a spatial resolution of 100 meters. Population density data were acquired through GEE scripts.

Before conducting the formal data analysis, the NPP data underwent rigorous preprocessing and validation. Invalid values from non-vegetation areas, such as water bodies and glaciers, were removed using masking techniques [29]. Additionally, unreasonable extreme values were corrected through statistical methods and spatial analysis [30]. This process significantly improved the accuracy and reliability of the data, laying a solid foundation for an in-depth study of the spatio-temporal variations in NPP across the Tibetan Plateau [14].

 

**Table 1. Factors influencing vegetation NPP change on the Tibetan Plateau.**

| Type | Factors | Indicator | Unit | Resolution | Classification method |
|---|---|---|---|---|---|
| Natural factors | X1 | Elevation | m | 100 m | Optimaldiscretisation |
| | X2 | Slope | ° | 100 m | Optimal discretisation |
| | X3 | Aspect | ° | 100 m | Optimal discretisation |
| | X4 | Soil moisture | m³/m³ | 1 km | Optimal discretisation |
| | X5 | Annual mean temperature | mm | 1 km | Optimal discretisation |
| | X6 | Annual mean precipitation | mm | 1 km | Optimal discretisation |
| | X7 | Accumulated temperature ≥0°C | ° | 1 km | Optimal discretisation |
| | X8 | Evapotranspiration | mm | 500 m | Optimal discretisation |
| | X9 | Total solar radiation | W/m² | 1000 m | Optimal discretisation |
| Human activities | X10 | Population density | people/km² | 100 m | Optimal discretisation |
| | X11 | GDP density | yuan/km² | 1000 m | Optimal discretisation |
| | X12 | Land use intensity | – | 1000 m | Optimal discretisation |
| | X13 | Distance to roads | m | 1000 m | Optimal discretisation |
| | X14 | Road density | m/km² | 1000 m | Optimal discretisation |
| | X15 | Distance to cropland | m | 1000 m | Optimal discretisation |
| | X16 | Distance to forestland | m | 1000 m | Optimal discretisation |
| | X17 | Distance to grassland | m | 1000 m | Optimal discretisation |
| | X18 | Distance to water bodies | m | 1000 m | Optimal discretisation |
| | X19 | River network density | m/km² | 1000 m | Optimal discretisation |
| | X20 | Distance to built-up areas | m | 1000 m | Optimal discretisation |

*Note*: This table summarises the key natural and human activity factors influencing the changes in NPP on the Tibetan Plateau, including their respective indicators, units, resolutions, variable types, and classification methods.

To categorize the natural and human activity data, an optimal discretization approach utilizing the Geodetector tool was implemented. Furthermore, to maintain consistency across the datasets and facilitate subsequent analyses, all data were projected using the Albers_Conic_Equal_Area projection, with the central meridian positioned at 105°E and the standard parallels set at 25°N and 47°N. The resolution for all variables was standardized to 500 meters.

## Research methods

Satellite data on NPP spanning from 2001 to 2021 were acquired and processed utilizing the Google Earth Engine (GEE) platform. To discern the spatial and temporal patterns of NPP changes, several non-parametric trend analysis techniques were employed, including the Sen slope estimator, Mann-Kendall test, coefficient of variation, and Hurst exponent. Following this, the OPGD model was used to investigate the interactive influences of natural factors and human activities on NPP distribution. This methodology enabled the identification of primary drivers and their optimal ranges for enhancing NPP growth, providing crucial insights for ecosystem management on the Tibetan Plateau. The technical approach of this study is depicted in Fig 2.

**Data processing using GEE.** Access the Google Earth Engine (GEE) platform via its website (https://earthengine.google.com/) and create a new script in the code editor [31,32,33]. Input the code necessary for downloading and processing data related to vegetation NPP, evapotranspiration, total solar radiation, and population density. After executing the script, download the datasets for vegetation NPP, evapotranspiration, total solar radiation, and population density over the Tibetan Plateau for the period from 2001 to 2021, all at a resolution of 500 meters. Subsequently, calculate the trends in vegetation NPP and conduct trend tests. The results will be saved as GeoTIFF files and exported to a designated folder

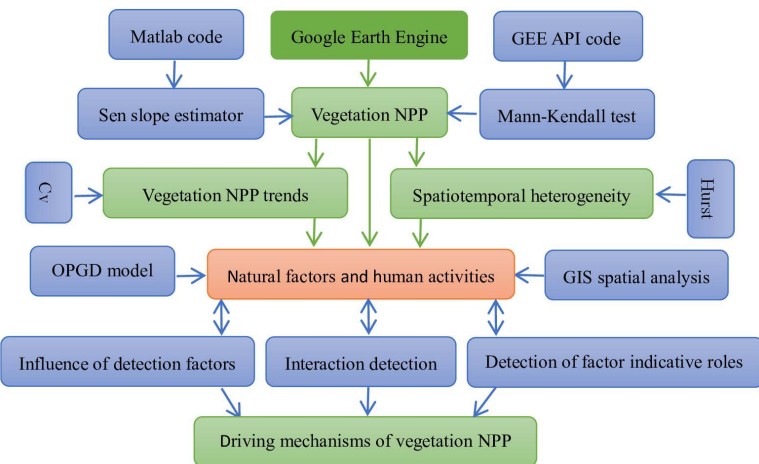

**Fig 2. The technical route of this study.**

in Google Drive [33]. Comprehensive details regarding the calculation procedures can be found in the supplementary materials.

**Coefficient of variation (Cv).** The coefficient of variation ($Cv$), also referred to as the coefficient of dispersion, is a key metric used to evaluate the stability and variability of data over time or across space [34,35]. As a dimensionless measure, $Cv$ facilitates comparisons without needing to reference the mean value of the dataset. Consequently, when assessing the stability of vegetation NPP, the ratio of the standard deviation to the mean serves as an effective comparison [36]. The formula for calculating $Cv$ is as follows:

$$Cv = \frac{\sigma}{u} \times 100\%$$

(1)

where, $Cv$ is the coefficient of variation for vegetation NPP, $\sigma$ is the standard deviation of vegetation NPP, representing the degree of data dispersion, and $u$ is the mean NDVI, representing the average vegetation NPP over a certain period or space. A higher $Cv$ indicates greater variability and less stability in vegetation NPP, whereas a lower $Cv$ suggests that vegetation NPP is relatively stable.

**Hurst index.** The Hurst index, a statistical measure of long-term dependence in time series data, is applied in this study to evaluate the persistence or anti-persistence of vegetation NPP trends over time. Following the method proposed by Hurst [37] and refined in subsequent studies,we compute the Hurst index using the NPP time series data [38,39,40]. The process involves calculating the cumulative deviation series, range, and standard deviation, followed by estimating the ratio $R(T)/S(T) \cong R/S$ and fitting it to the scaling relationship $R/S \propto T^H$, the calculation of the Hurst index are as follows:

(1) Given Time Series.

Let the time series be {NPP(t)}, where t = 1, 2, …, n.

(2) Compute the mean.

The mean of the time series is calculated as:

$$\overline{NPP_{(T)}} = \frac{1}{T} \sum_{t=1}^{T} NPP_{(T)} \quad T = 1, 2, ..., n$$

(2)

(3) Construct the cumulative deviation series

The cumulative deviation series $X_{(x,T)}$ represents the cumulative deviation of the series from its mean:

$$X_{(t,T)} = \sum_{t=1}^{T} (NPP_{(t)} - \overline{NPP_{(T)}})\, 1 \leq t \leq T \tag{3}$$

(4) Calculate the range series:

The range $R_{(T)}$ is the difference between the maximum and minimum of the cumulative deviation series:

$$R_{(T)} = \max\, X_{(t,T)} - \min\, x_{(t,T)}\, T = 1, 2, ..., n \tag{4}$$

(5) Calculate the standard deviation series:

The standard deviation $S_{(T)}$ of the original time series is computed as:

$$S_{(T)} = \left[ \frac{1}{T} \sum_{t=1}^{T} (\text{NPP}_{(t)} - NPP_{(T)}2) \right]^{\frac{1}{2}} T = 1, 2, ..., n \tag{5}$$

(6) Calculate the Hurst index:

The Hurst index $H$ is estimated using the following relationship:

$$H = \frac{\log(R/S)_n - a}{\log(n)} \tag{6}$$

This allows us to classify the time series into three categories: anti-persistent ($0<H<0.5$), random ($H=0.5$), and persistent ($0.5<H<1$). Through this analysis, the study quantifies the degree of persistence in NPP fluctuations, offering critical insights into how vegetation dynamics respond to environmental and anthropogenic influences, particularly under the context of climate change [41,42].

**Trend analysis.** The Sen's slope estimator and Mann-Kendall trend test are non-parametric statistical methods extensively applied to analyze trends in time series data, first introduced by Hirsch and Slack [43]. In recent years, studies have increasingly combined these techniques with other methods to enhance the precision of trend assessments. For instance, research by De Jong et al.[44]and Gocic and Trajkovic highlights the effectiveness of integrating these approaches in uncovering trends within time series datasets [45]. Owing to their complementary strengths, these methods have gained broad applications in remote sensing data analysis. In this study, these methods are employed to detect the spatiotemporal variations in NPP across the research area, providing robust trend estimates for long-term changes in vegetation productivity. **Sen's slope estimation**. Sen's slope estimation is a reliable non-parametric approach for identifying trends in time series datasets [45,46]. In this study, the method is applied by first arranging observed NPP values in chronological order, followed by dividing the data into several groups. The slopes between all data points within each group are calculated, and the median slope is selected as the group's trend estimate. The overall NPP trend is determined by calculating the median slope across all groups [47]. This approach reduces the influence of outliers and extreme values, which is critical for ensuring accurate trend detection in large-scale NPP time series datasets [48].

The formula for calculating Sen's trend estimation is as follows:

$$\beta = median \frac{NPP_j - NPP_i}{j - i}, \ 2000\, i \leq j \leq 2020 \tag{7}$$

where, $\beta$ represents the trend in vegetation NPP, and $i$ and $j$ refer to the indices of the time series. $NPP_i$ and $NPP_j$ are the values of vegetation NPP at time series $i$ and $j$, respectively. When $\beta>0$, it indicates an increasing trend in vegetation NPP, whereas $\beta<0$, reflects a decreasing trend. The larger the absolute value of the slope, the more significant the change in vegetation NPP [45,47].

**Mann-Kendall trend test**. The Mann-Kendall trend test is a commonly used non-parametric method for evaluating the significance of trends in time series data [49,50,51,52]. This method has been extensively applied in ecosystem NPP studies, particularly for assessing long-term impacts of climate change and land-use changes on NPP [53,54]. In this study, the Mann-Kendall trend test is used to evaluate the significance of spatiotemporal changes in vegetation NPP, enabling a comprehensive understanding of trend dynamics and their underlying drivers.

The calculation formula for the Mann-Kendall trend test is as follows:

$$S = \sum_{j=1}^{n-1} \sum_{i=j+1}^{n} (NPP_j - NPP_i) \tag{8}$$

$$sng = (NPP_j - NPP_i) = \begin{cases} 1, NPP_j - NPP_i > 0 \\ 0, NPP_j - NPP_i = 0 \\ -1, NPP_j - NPP_i < 0 \end{cases} \tag{9}$$

$$Z = \begin{cases} \frac{S}{\sqrt{Var(S)}}, S > 0 \\ 0, S = 0 \\ \frac{S+1}{\sqrt{Var(S)}}, S < 0 \end{cases} \tag{10}$$

$$Var(S) = \frac{n(n-1)\,(2n+5)}{18} \tag{11}$$

Where, $NPP_i$ and $NPP_j$ represent the NPP values for a pixel in year $i$ and year $j$, respectively, with $n$ denoting the length of the time series, which spans 21 years. The standardized normal statistic is represented by $Z$, and $S$ is the trend statistic, which approximately follows a normal distribution, calculated using NPP values for the period from 2001 to 2021. The function $sng$ is the sign function, and $Var(S)$ refers to the variance of the statistic. This study uses a two-tailed test with a significance level $\alpha$=0.05, which yields a critical value of $Z_1-\alpha/2$= ±1.96. When the absolute value of the $Z$-statistic exceeds 1.65, 1.96, or 2.58, the trend is significant at the 90%, 95%, and 99% confidence levels, respectively [14,15].

**OPGD.** The Optimal Parameters-based Geographical Detector (OPGD) model is a statistical method designed to assess spatial heterogeneity and its driving factors. By optimizing parameter selection, the OPGD model improves upon the traditional Geographical Detector (GD) model, allowing for more accurate quantification of the influence of both natural and human activities on spatial patterns.

**Parameter optimization.** The OPGD model utilizes optimization algorithms that adjust the detector's parameters automatically according to the characteristics of each variable, ensuring the model's suitability for complex spatial analysis [55]. For factor detection, this study employs an optimized discretization approach, calculating the Q value through combinations of breakpoints for continuous variables. The factor detector assesses different classification methods and breakpoint combinations, selecting the one that maximizes the Q value as the optimal parameter set [55]. This method enhances the identification of spatial stratified heterogeneity and highlights the influence of key continuous variables on spatial variation [55,56,57]. By choosing the parameter set with the highest Q value, the model accurately detects spatial heterogeneity in vegetation NPP on the Tibetan Plateau and identifies the factors driving these changes [58]. Optimizing the discretization parameters increases the explanatory power of continuous geographical variables, offering deeper insights into the spatial distribution and dynamics of NPP [55].

**Spatial differentiation analysis.** The OPGD model assesses the importance of each variable by comparing the variance between the observed values in the study area and the variable layers under different parameter settings[55,59]. It calculates the Q value to determine the explanatory power of each factor in relation to spatial heterogeneity, identifying the primary drivers[55]. The process involves the following steps: First, the vegetation NPP layers are overlaid with variable factor layers to uncover their spatial relationships. This helps in understanding the association between vegetation NPP and various factors. Second, the spatial classification of different factors is performed by grouping them into regions or categories, providing a clearer view of the factors' impact across different spatial scales. Finally, a significance test is conducted to compare the mean differences between various factors, which helps determine their contribution to changes in vegetation NPP and identify the most influential factors driving NPP variations.

The explanatory power (Q) of the factors is calculated using the following model:

$$Q = 1 - \left( \sum_{h=1}^{L} N_h \sigma_h^2 \right) / (N\sigma^2) = 1 - \frac{SSW}{SST} \tag{12}$$

where, Q represents the explanatory power of the factor on vegetation NPP, with a range of [0, 1]; the larger the value, the stronger the factor's explanatory power for vegetation NPP. $h = 1,..., L$ denotes the layers of vegetation NPP and influencing factors. $N_h$ is the number of units in layer $h$. $N$ is the total number of units in the region. $\sigma_h^2$ and $\sigma^2$ are the variances of vegetation NPP in layer $h$ and the entire region, respectively. $SSW$ represents the sum of the within-layer variances. $SST$ is the total variance across the entire region.

The variance of the regional Y value is calculated as follows:

$$\sigma^2 = \frac{1}{N-1} \sum_{i=1}^{N} \left( Y_i - \overline{Y} \right)^2 \tag{13}$$

where, $Y_j$ and $\overline{Y}$ represents the value of sample $j$ and is the mean value of $Y$ for the entire region.

**Multifactor interaction analysis.** The OPGD model allows for the analysis of interactions between natural and human factors, shedding light on their combined effects on spatial phenomena. By exploring these interactions, the model helps to clarify the complex relationships between various driving forces and identifies the key factors influencing NPP. The factor detector calculates the relative importance of each variable and reveals how overlapping spatial variables interact. Interaction effects emerge when two variables are spatially combined, reflecting their collective impact [55].

The interaction detector compares the Q value of the combined variables to the individual Q values, determining whether the interaction is enhancing, weakening, or independent. Five interaction types are identified: nonlinear weakening, single-variable weakening, double-variable enhancement, independence, and nonlinear enhancement [55,59].

**Risk detection.** Risk detection analyzes how a specific factor (such as environmental or socio-economic variables) influences the distribution of a spatial phenomenon (e.g., disease, pollution, natural disasters) [55,59]. The method compares the factor's distribution across regions with the target phenomenon, determining whether the factor significantly affects spatial heterogeneity. The *t*-statistic is used to test whether the mean NPP in subregion $h$ differs significantly from the overall mean:

$$t = \frac{\overline{Y}_{h=1} - \overline{Y}_{h=2}}{\left[ \frac{Var(Y_{h=1})}{n_{h=1}} + \frac{Var(Y_{h=2})}{n_{h=2}} \right]^{1/2}} \tag{14}$$

where, $\overline{Y}_h$ represents the mean NPP attribute value within subregion $h$, $n_h$ is the number of samples within subregion $h$, $Var$ denotes the variance of the NPP values. This formula assesses whether the mean NPP in subregion $h$ significantly differs from the overall region's mean.

**Ecological detection.** Ecological detection compares the influence of different ecological factors on the spatial distribution of NPP, determining whether certain variables have a more pronounced effect [55,59]. The *F*-statistic is employed to evaluate whether significant differences exist between the impacts of these factors on NPP distribution:

$$F = \frac{N_{x1} \times (N_{x2}-1) \times SSW_{x1}}{N_{x2} \times (N_{x1}-1) \times SSW_{x2}} \tag{15}$$

$$SSW_{x1} = \sum_{h=1}^{L_1} N_h \sigma_h^2 \tag{16}$$

$$SSW_{x2} = \sum_{h=1}^{L_2} N_h \sigma_h^2 \tag{17}$$

where, $N_{x1}$ and $N_{x2}$ are the sample sizes of two factors, and $SSW_{x1}$ and $SSW_{x2}$ represent the sums of intra-layer variances. $L_1$ and $L_2$ indicate the number of stratifications for variables $x_1$ and $x_2$, respectively.

## Results and analysis

### Temporal variation characteristics

From 2001 to 2021, the average NPP of vegetation on the Qinghai-Tibet Plateau exhibited a fluctuating trend (Fig 1D). Spatial statistical analysis indicates that the average NPP generally showed a trend of fluctuating increase, particularly from 2005 to 2021, during which the annual average NPP grew significantly. The highest NPP value was recorded in 2021, reaching 208.53 gC m$^{-2}$.a$^{-1}$, while the lowest value appeared in 2004, at 177.90 gC m$^{-2}$.a$^{-1}$. Meanwhile, the fluctuations in the average NPP were quite pronounced, especially in certain years such as 2006 and 2013, when NPP values were relatively high, whereas lower values were observed in 2012 and 2014 (Fig 1). These fluctuations may be closely related to interannual climate variations, changes in precipitation and temperature, as well as the impacts of human activities such as grazing and land use [15].

### Spatial pattern characteristics

**Spatial distribution.** The spatial distribution of vegetation NPP across the Qinghai-Tibet Plateau reveals pronounced regional disparities, varying significantly between different areas and over time. Higher NPP values are predominantly observed in the southeastern regions of the plateau, while lower values are concentrated in the northwest. Favorable climatic conditions in the eastern and southern regions, including increased humidity and denser vegetation, contribute to relatively high NPP levels. In contrast, the drier climates and higher altitudes in the western and northern areas result in significantly lower NPP, with some regions displaying values close to zero (Fig 3).

**Temporal trends in NPP distribution.** As shown in Fig 3, the spatial patterns of NPP exhibit notable changes over the years: In 2001, the maximum NPP value was 1905 gC m$^{-2}$·a$^{-1}$, primarily concentrated in the southeastern portion of the plateau, reflecting robust vegetation growth in this region (Fig 3A). By 2011, the maximum NPP had increased to 1947 gC m$^{-2}$·a$^{-1}$, indicating improved vegetation productivity, particularly in the southeast [20,60] (Fig 3B). This increase is likely associated with favorable climatic changes, such as higher precipitation and warmer temperatures, which fostered an expansion of vegetated areas [9,60].However, by 2021, the peak NPP value had declined to 1731 gC m$^{-2}$·a$^{-1}$, suggesting reduced productivity over the preceding decade. This decline may be attributed to adverse factors such as climate change, leading to droughts and extreme weather events, as well as land use changes like overgrazing and urban development, which have likely suppressed vegetation growth (Fig 3C). From 2001 to 2021, the highest NPP values

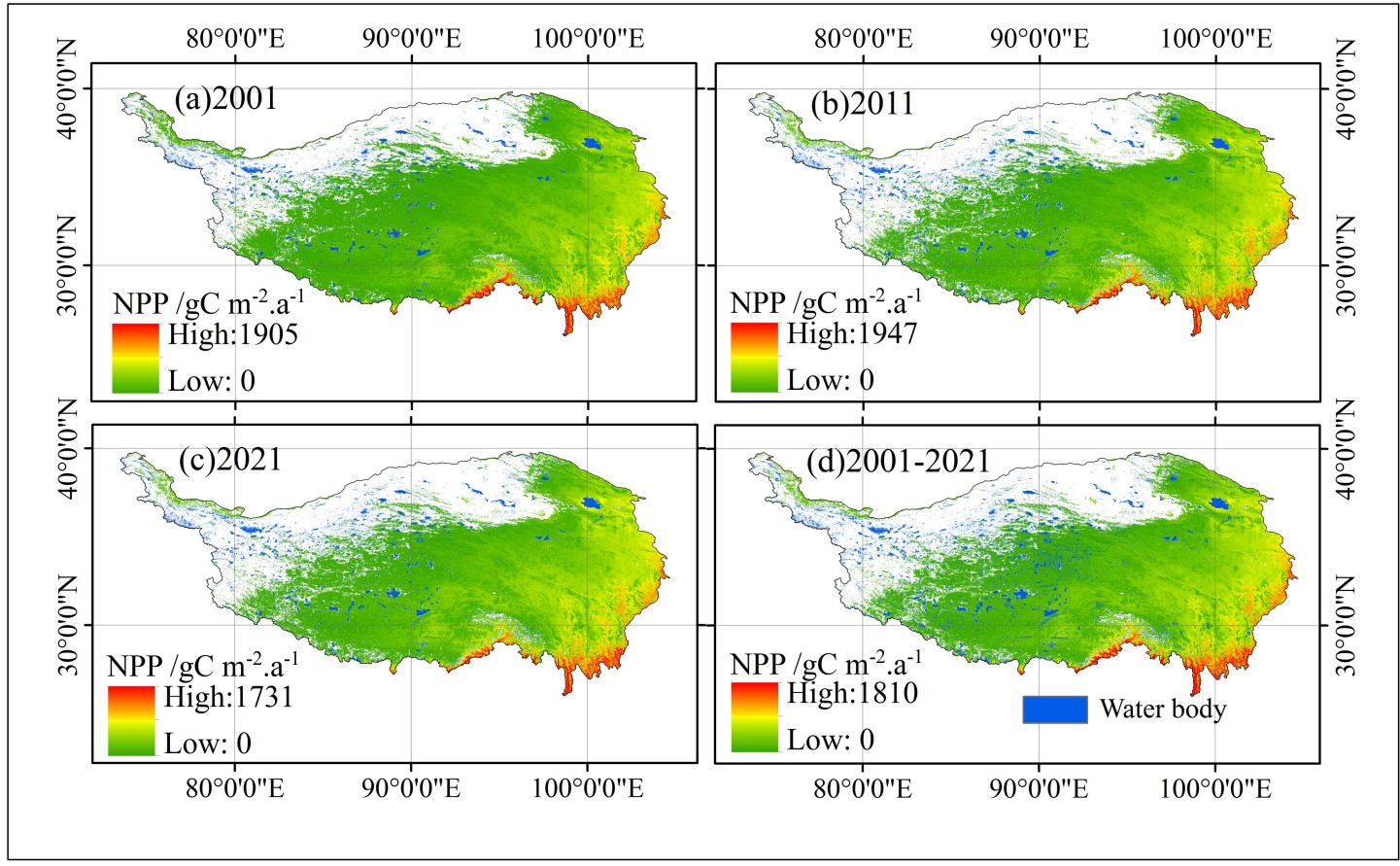

**Fig 3. Vegetation NPP spatial pattern changes in the Qinghai-Tibet Plateau from 2001 to 2021.**

consistently occurred in the southeastern part of the plateau, particularly near the Hengduan Mountains and the Sichuan Basin. Over the study period, the maximum recorded NPP in this region reached 1810 gC m$^{-2}$·a$^{-1}$, with an average of 724.72 gC m$^{-2}$·a$^{-1}$ (Fig 3D).

**Regional distribution.** The central and northwestern regions exhibited significantly lower NPP values, averaging 34.21 gC m$^{-2}$·a$^{-1}$, particularly in the arid and alpine zones of the northwest. Southeastern lowlands supported abundant vegetation and high productivity, while the sparse or absent vegetation in the high-altitude interior and northern regions resulted in much lower NPP levels.

In summary, the distribution of NPP on the Qinghai-Tibet Plateau exhibits a distinct zonal pattern [20]. The southeastern fringes maintain higher productivity due to favorable climatic and environmental conditions, whereas the interior regions, characterized by harsh climates and sparse vegetation, display substantially lower NPP values (Fig. 3D).

## Vegetation NPP variation in different climate zones

Using the Chinese climate zoning map compiled by the China National Meteorological Administration (1978), based on climate data from 1951 to 1970, and combining this with spatial statistical analysis of vegetation NPP on the Qinghai-Tibet Plateau from 2001 to 2021, the NPP values for different climate zones were derived (Table 2, Fig 4). This analysis revealed significant differences in vegetation NPP across various climate zones on the plateau. Furthermore, the

**Table 2. Statistics of climate zones and NPP on the Qinghai-Tibet Plateau from 2001 to 2021 (unit: gC m$^{-2}$.a$^{-1}$).**

| Secondary zoning code | Name | Mean | STD |
|---|---|---|---|
| IIC5 | Yining Region | 94.85 | 38.50 |
| IIID1 | Southern Xinjiang Region | 109.14 | 9.61 |
| HD2 | Northern Tibet Region | 50.10 | 30.28 |
| HC3 | Southern Tibet Region | 117.34 | 106.05 |
| HC2 | Central Tibet Region | 86.69 | 51.33 |
| IIIB3 | Weihe Region | 552.99 | 129.06 |
| IVA2 | Qinba Region | 641.13 | 186.27 |
| VA3 | Sichuan Region | 517.28 | 227.78 |
| HB2 | Chamdo Region | 225.03 | 93.68 |
| HA1 | Bomi-Western Sichuan Region | 342.50 | 173.76 |
| HV1 | Dawang-Zayu Region | 551.48 | 327.60 |
| HC1 | Qilian-Qinghai Lake Region | 211.03 | 120.60 |
| HB1 | Southern Qinghai Region | 132.69 | 69.29 |
| HD1 | Qaidam Basin Region | 132.96 | 62.18 |
| IIC2 | Central Mongolia Region | 361.98 | 124.20 |
| VA5 | Northern Yunnan Region | 784.65 | 260.93 |

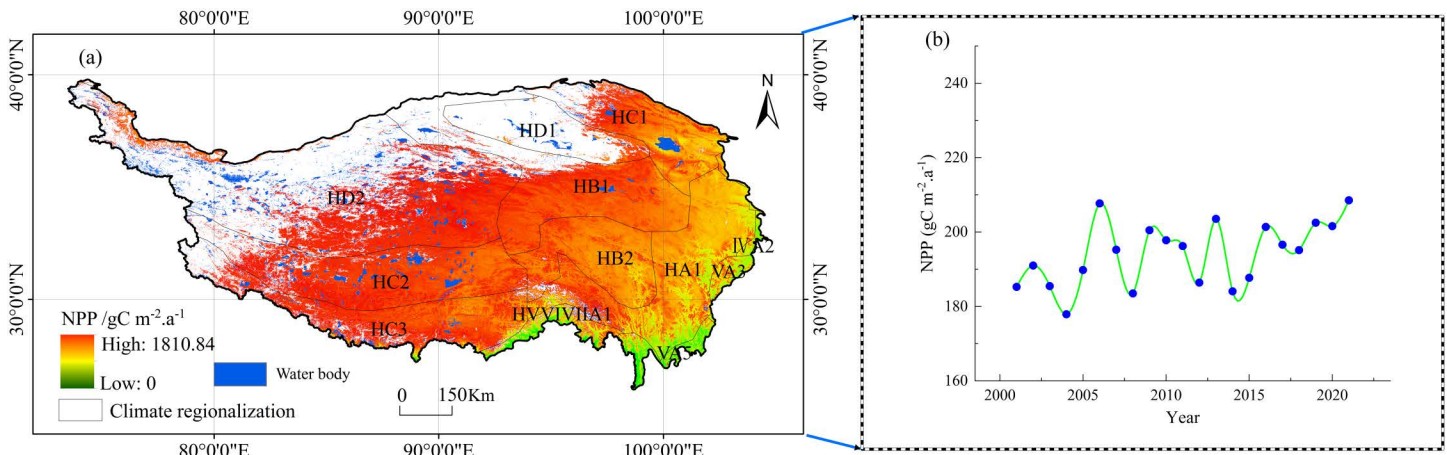

**Fig 4. Climatic zoning and NPP changes on the Qinghai-Tibet Plateau.** (A) Climatic zoning codes and average vegetation NPP from 2001 to 2021. (B) annual NPP variation on the Qinghai-Tibet Plateau. The classification is based on climatic zoning codes, with the climate codes shown in the figure corresponding to those listed in Table 2.

relationship between NPP mean and variance highlights the combined effects of climate and topography on vegetation productivity [61].

**The role of topography and climate in vegetation productivity.** Topographic complexity plays a critical role in shaping vegetation productivity on the plateau by influencing climate, hydrological processes, and local environmental conditions [62]. Variations in topography lead to differences in factors such as water availability and climate change, resulting in spatial disparities in NPP [63]. These interactions vary across regions, underscoring the importance of understanding the mechanisms driving spatiotemporal changes in vegetation productivity [61].

NPP distribution across climate zones. In the Central and Northern Subtropical climate zones, vegetation NPP values are significantly higher compared to other zones. For instance, the Northern Yunnan and Qinba regions exhibit mean NPP values of 784.65 gC $m^{-2} \cdot a^{-1}$ and 641.13 gC $m^{-2} \cdot a^{-1}$, respectively. These areas benefit from warm and humid climatic conditions, which support stronger vegetation productivity.

Conversely, in the Plateau climate zones, such as the Northern and Central Tibetan regions, NPP values are much lower, with averages of 50.10 gC $m^{-2} \cdot a^{-1}$ and 86.69 gC $m^{-2} \cdot a^{-1}$, respectively. This reflects the limitations imposed by cold climates and high altitudes on vegetation growth (Table 2, Fig 4).

**Spatial variability of vegetation NPP.** Certain regions, such as Dawang-Zayu, Northern Yunnan, and Sichuan, exhibit high NPP variability, with standard deviations of 327.60, 260.93, and 227.78, respectively (Table 2, Fig. 4). These variations likely arise from complex terrain, diverse climatic conditions, or human activities.

In regions like Qinba and Bomi-Western Sichuan, NPP variability is moderate, with standard deviations of 186.27 and 173.76, respectively. This indicates that while spatial differences exist, vegetation productivity changes are relatively controlled. Regions such as Southern Xinjiang display low variability, with a standard deviation of only 9.61. This suggests that vegetation productivity in these areas is relatively stable, likely due to consistent climatic conditions and homogeneous vegetation types (Table 2, Fig. 4).

**Correlation between NPP mean and variance.** There is a notable correlation between NPP mean and variance across regions. Areas with higher NPP, such as Northern Yunnan, Dawang-Zayu, and Sichuan, tend to exhibit greater variability, indicating a stronger influence of environmental factors such as terrain, precipitation, and human activities [8,54] (Table 2, Fig 4). On the other hand, regions with low NPP, such as Northern Tibet and Southern Xinjiang, show lower variability. This suggests that vegetation productivity in these areas is more constrained by limiting factors, leading to reduced fluctuations (Table 2, Fig 4).

## Spatial fluctuation, change trend, and trend analysis

**Spatial fluctuation.** The spatial variability of vegetation NPP on the Qinghai-Tibet Plateau, expressed as the coefficient of variation (Cv), ranges from 0 to 4.58 gC $m^{-2} \cdot a^{-1}$ (Fig 5A). Based on the classification schemes proposed in previous studies (Peng et al., 2014; [5,6]), Cv is divided into five fluctuation levels: low (0–0.05 gC $m^{-2} \cdot a^{-1}$), relatively low (0.05–0.13 gC $m^{-2} \cdot a^{-1}$), moderate (0.13–0.23 gC $m^{-2} \cdot a^{-1}$), relatively high (0.23–1.03 gC$m^{-2} \cdot a^{-1}$), and high (1.03–4.58 gC $m^{-2} \cdot a^{-1}$) (Fig. 5A).

Spatially, low and relatively low fluctuation areas dominate the plateau, covering about 73.78% of its total area. These zones are primarily located in the arid northern desert regions. Despite their sparse vegetation, these regions exhibit minimal NPP variability, likely due to the inherent resilience of the vegetation to harsh environmental conditions.

Moderate fluctuation zones, accounting for 23.77% of the plateau's area, are more widely distributed, occurring across the northeast, central, southeast, and southwest regions. These areas show increased NPP variability, which may result from climatic fluctuations and anthropogenic influences.

High and relatively high fluctuation zones are limited, comprising only 2.45% of the plateau's area. These regions are predominantly situated in the southeast, where factors such as complex topography and intensified human activities contribute to the higher variability in NPP.

In summary, the Qinghai-Tibet Plateau displays notable spatial variability in vegetation NPP. Most of the plateau exhibits stable NPP dynamics, while only a small fraction experiences significant fluctuations due to environmental and human-induced factors.

**Change trend analysis.** From 2001 to 2021, the Hurst index of vegetation NPP on the Qinghai-Tibet Plateau ranged from -0.08 to 0.96, with a mean value of 0.41 and a standard deviation of 0.08. To interpret these values, the index is divided into five categories: <0 (-0.08 to 0), 0–0.3, 0.3–0.5, 0.5–0.7, and >0.7 (0.7–0.96) (Fig. 4B).

The spatial distribution of the Hurst index reveals chaotic patterns with no evident clustering, reflecting the heterogeneity of vegetation dynamics across the plateau. Regions where $H<0.5$ account for 40.63% of the area, suggesting that the

NPP time series in these zones are anti-persistent, meaning that future trends are likely to reverse previous patterns. In contrast, areas with *H* >0.5 comprise 8.80%, indicating persistence in NPP trends, where future changes are expected to follow historical trajectories. The remaining 50.57% of the plateau exhibits *H* =0.5, signifying random NPP dynamics with no clear tendency for persistence or anti-persistence, making future trends difficult to predict.

These findings highlight the complexity and variability of vegetation changes across the Qinghai-Tibet Plateau. The significant proportion of anti-persistent and random regions suggests that future NPP trends may be highly dynamic, with potential reversals and uncertainties dominating the plateau's vegetation development.

**Sen trend and MK test.** The fitted slope of vegetation NPP on the Qinghai-Tibet Plateau ranges from −423.93 to 299.36 gC m$^{-2}$.a$^{-1}$, with 62.93% of the area showing a positive slope, indicating an overall increasing trend in NPP (Fig 5C). Significant increases or stability in NPP are observed in the northeastern, central, eastern, and southeastern regions of the Qinghai-Tibet Plateau, whereas declines are seen in the southwestern, southern, and parts of the eastern regions.

According to Fig 5D and Table 3, most areas of the Qinghai-Tibet Plateau show stable or slightly increasing NPP trends, with significantly decreasing regions being relatively scarce. While NPP has decreased in some areas, the over-all trend indicates ecosystem stability or improvement, particularly with prominent increases in the northern and eastern regions. Areas with significant increases in vegetation NPP (fitted slope greater than 2.54 gC m$^{-2}$.a$^{-1}$, Z-value greater than 1.96, *p*<0.01 or 0.01≤*p*≤0.05) are marked in dark green (*β* = 2), covering approximately 707,963 km², or 27.85% of the

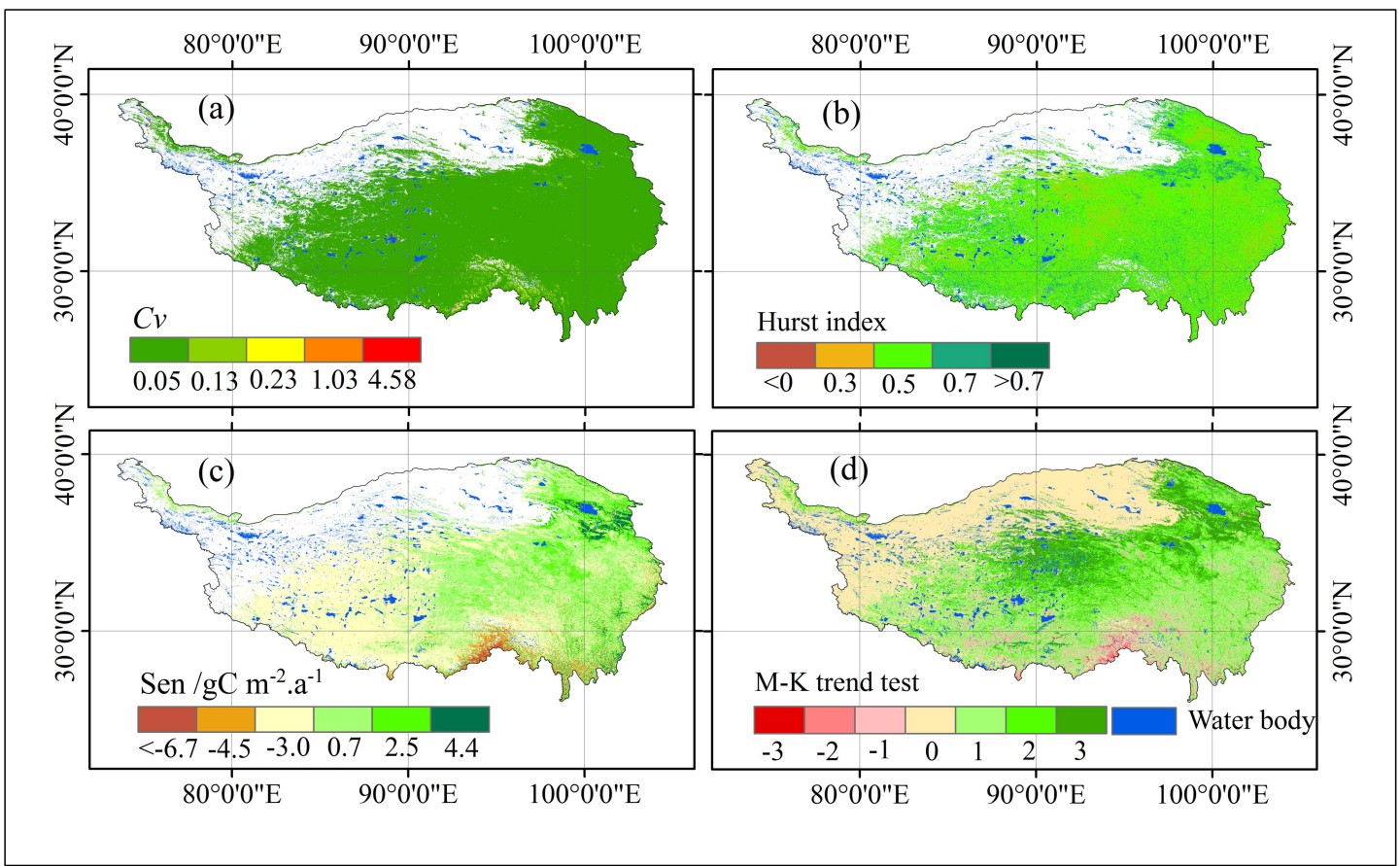

**Fig 5.** *Cv* **(A), Hurst index (B), Sen (C), and significance test (D) of vegetation NPP on the Qinghai-Tibet Plateau from 2001 to 2021.** In D, the scale from -3 to 3 represents the classification of the NPP trend significance test, corresponding to the categories outlined in Table 3.

total area, mainly concentrated in the northern, western, and southern margins. This may reflect the impact of ecological restoration measures and the extended growing season due to climate warming.

Areas of sharp NPP decline (fitted slope less than −4.48 gC m$^{-2}$.a$^{-1}$, Z-value less than −1.96, $p<0.01$ or $0.01≤ p <0.05$) are marked in red ($β = −2$), primarily located in the eastern and southern parts of the plateau (Table 3, Fig 3D), where human activity is frequent. The decline in NPP in these areas could be related to human exploitation, climate change, and vegetation degradation.

Regions with no significant change in NPP (fitted slope between −2.99 and 0.68 gC m$^{-2}$.a$^{-1}$, with $−1.65 ≤ Z ≤ 1.65$, $p ≥ 0.05$) are shown in yellow. These areas cover the largest portion, with an area of 784,630 km², accounting for 30.86% of the total, mainly distributed in the central and northern parts of the plateau (Table 3, Fig 5D), likely due to the limited number of NPP pixels and insignificant changes.

Slightly increasing NPP areas (fitted slope between 0.69 and 2.54 gC m$^{-2}$.a$^{-1}$, $1.65< Z <1.96$, and $p≥0.05$) and slightly decreasing areas (fitted slope between −4.48 and −2.99 gC m$^{-2}$.a$^{-1}$, $−1.65< Z <-1.96$, $p≥0.05$) are mainly concentrated in parts of the central and northern plateau and parts of the eastern and southern plateau, respectively (Table 3, Fig 5D). These regions are marked in orange ($β = −1$) and light green ($β = 1$), accounting for 51.55% and 1.32% of the total area, respectively, which may be associated with slight climate fluctuations or minimal human impacts.

### Analysis of drivers for vegetation NPP change

**Variables discretization.** The OPGD model in this study was applied to analyze the spatial explanatory variables associated with vegetation NPP changes. The model is capable of integrating both categorical and continuous variables,where categorical variables are incorporated directly into the geographical detector model. For continuous variables, discretization was performed through Q optimization, applying different grading methods to identify breakpoints and determine the best parameter combinations. The results reveal that the optimal parameters and number of breakpoints differ across various explanatory factors (Fig 6A, B).

Variables such as X1, X2, X7, X9, and X13 were classified into nine categories using the natural breaks method. X3 and X8, however, were divided into nine levels via the equal interval method. Meanwhile, variables like X4, X6, X14, X18, X19, and X20 were split using the standard deviation method, also into nine levels. Variables X10, X11, X12, and X16 were grouped according to the quantile method, again with nine levels. Lastly, X15 was categorized into eight levels and X17 into six levels, both based on the quantile method.

**Influence of detection factors.** The findings from the factor analysis reveal that the variables elevation, total solar radiation, annual average temperature, and accumulated temperature above 0 °C exhibit the highest Q values of 0.5696, 0.5634, 0.5367, and 0.5327, respectively. Each of these factors accounts for over 53% of the overall variation in NPP,

**Table 3. Statistical summary of vegetation NPP trend from 2001 to 2021.**

| β | Sen | Z value | Degree | Area/km² | Percentage/% |
|---|---|---|---|---|---|
| -2 | <-4.48 | $Z<-1.96$ , $p<0.01$ , $0.01≤p<0.05$ | Severely decrease** | 33649.7 | 1.32 |
| -1 | -4.48~ -2.99 | $-1.65<Z<-1.96$ , $p≥0.05$ | Slightly decrease | 213984 | 8.42 |
| 0 | -2.99~0.68 | $-1.65 ≤Z ≤ 1.65$ , $p ≥ 0.05$ | No change | 784630 | 30.86 |
| 1 | 0.68~2.54 | $1.65<Z<1.96$ , $p≥0.05$ | Slightly increase | 802122 | 31.55 |
| 2 | >2.54 | $Z>1.96$ , $0.01≤p<0.05$ , $p<0.01$ | Significantly increase** | 707963 | 27.85 |

*Note*: The significance test method used is the *t*-test, where * indicates that *p* passes the 0.05 confidence level test, and ** indicates that *p* passes the 0.01 confidence level test. For pixels where $-2.99 <Sen < 0.68$, $-1.65 ≤ Z ≤ 1.65$, and $p ≥ 0.05$, the number of vegetation NPP pixels is extremely small, thus these pixels are classified as no-change types.

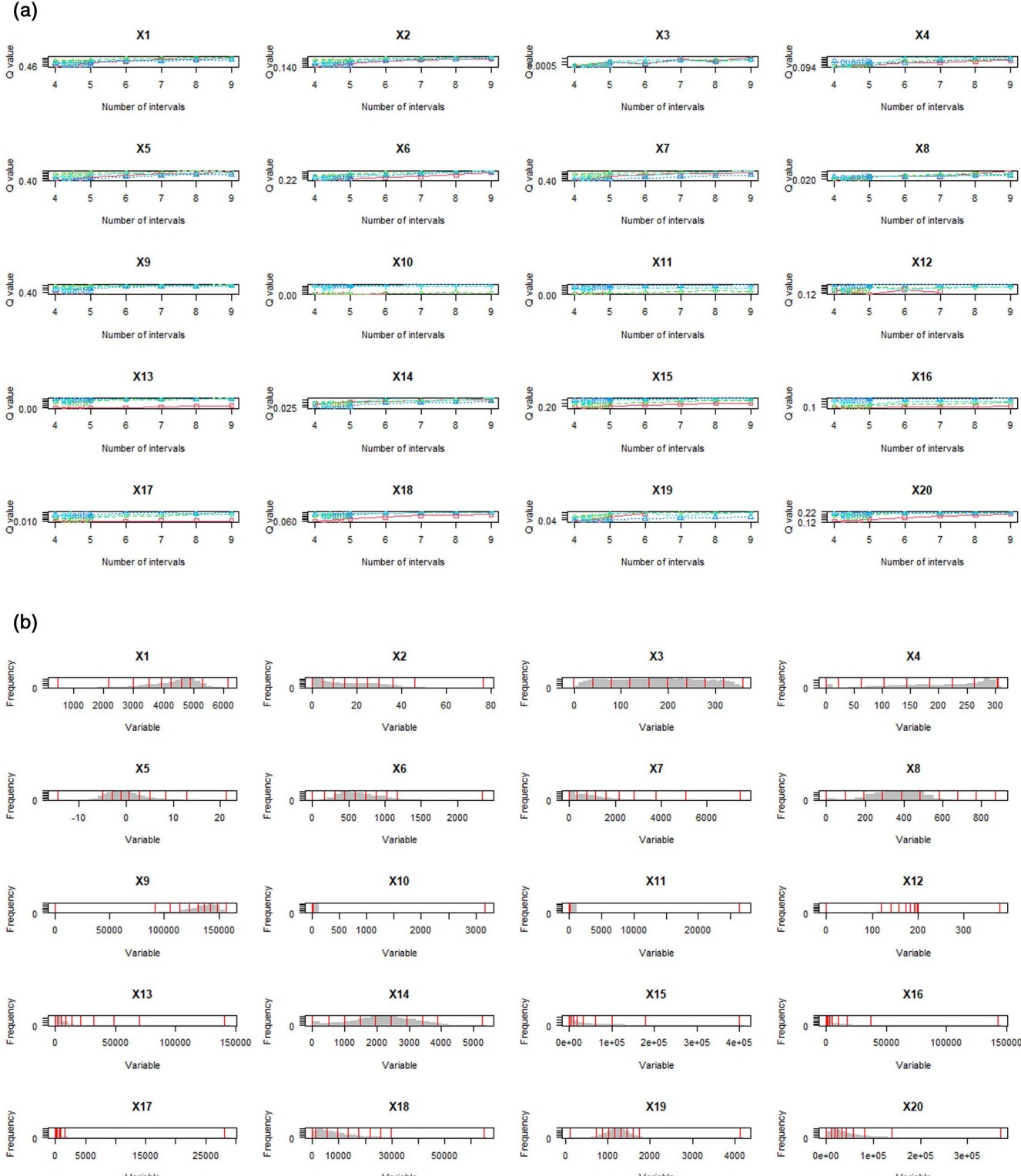

**Fig 6. Exploration of explanatory variables for vegetation NPP Change based on OPGD** : the process of spatial data discretization parameter optimization (A) and the results (B). The meanings of X1-X20 are provided in Table 1.

signifying their substantial influence (Fig 7A). This indicates that natural elements like elevation, solar radiation, and temperature are key contributors to changes in NPP, highlighting their dominant role.

Regarding human activity factors, forest distance and road distance show Q values of 0.4139 and 0.3254, respectively, both surpassing 32% in explanatory power, suggesting that these distances significantly impact NPP variability. Additionally, population density and GDP density demonstrate Q values of 0.2839 and 0.2740, also exceeding 27%, indicating a

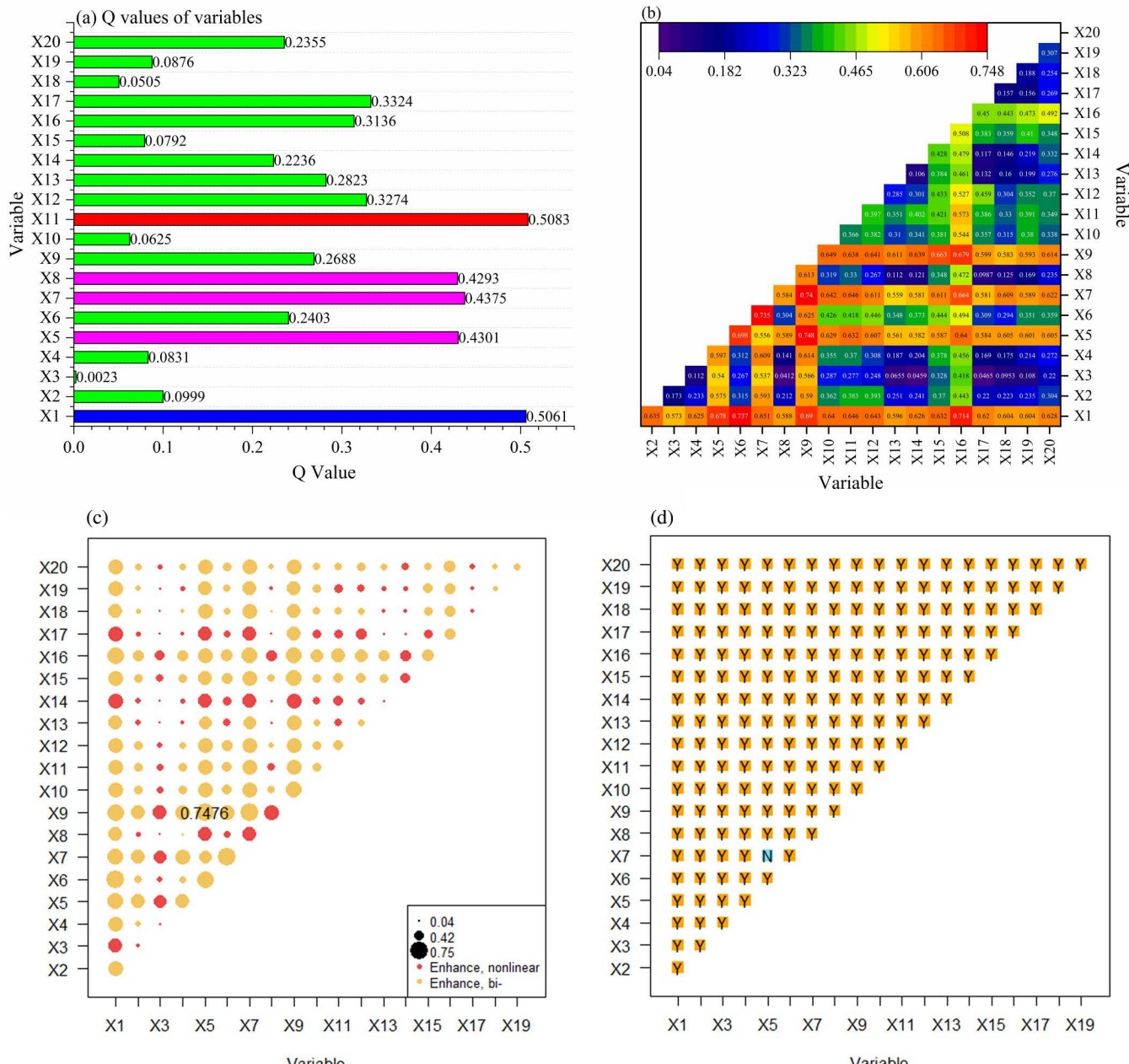

**Fig 7. Calculation results of the OPGD model.** (A) Q values of individual variables; (B) correlation coefficient matrix of Q value; (C) interaction detection results; (D) ecological detection outcomes.

meaningful effect. The Q values for annual mean precipitation and land use intensity are recorded at 0.2637 and 0.2462, respectively, suggesting that both natural and socio-economic factors contribute to NPP variation.

In contrast, the Q values for slope, aspect, soil moisture, evapotranspiration, distance to water bodies, and proximity to built-up areas are relatively low, measuring 0.1683, 0.0017, 0.1084, 0.0372, 0.1053, and 0.0925, respectively. These values indicate an explanatory power below 17%, suggesting that these factors have a minimal individual influence on NPP (Fig 7A).

In conclusion, key natural factors such as elevation, solar radiation, and annual mean temperature are essential to understanding NPP variations. Simultaneously, human activity factors including forest distance, road distance, population density, and GDP density play significant roles, while the impacts of slope, aspect, soil moisture, and evapotranspiration are comparatively limited.

**Interaction detection of factors.** This analysis employs interaction detection to assess how various natural and anthropogenic factors collectively influence NPP, determining whether they enhance, diminish, or act independently on this dependent variable (Fig 7B, C).

The size of the circles indicates the strength of the interactions, with red representing nonlinear enhancement and yellow representing bilinear enhancement in Fig 7C. The findings from Fig 6B and C reveal significant interactions among elevation, annual precipitation, annual temperature, and solar radiation.Notably, the interaction between elevation and annual precipitation is the most pronounced, indicating that precipitation at high altitudes significantly affects NPP. This combined effect exceeds the influence of each individual factor, highlighting the critical role of water and heat availability in vegetation growth at elevated locations. For instance, the interaction strengths rank as follows: X1∩X6 (0.7373) > X1∩X5 (0.6783) > X1∩X11 (0.6458).

In addition, the interplay between population density and GDP, along with land use intensity, shows significant effects, particularly the interaction between population density and GDP. This finding suggests that in economically developed and densely populated regions, human activities substantially impact NPP, especially when land use changes occur, amplifying the overall influence of human activities. The interaction strengths for this pairing are as follows: X12∩X13 (0.6116) > X12∩X14 (0.5332) (Fig 7B, C). Moreover, the relationship between road distance and both forest distance and cropland distance exhibits enhancement effects, particularly highlighted by the interaction of road distance and forest distance. This relationship emphasizes how road construction's spatial dynamics with forest areas critically influence NPP, likely due to deforestation impacts from road development. The interaction strengths are illustrated as X15∩X17 (0.5284) > X15∩X16 (0.5107). Furthermore, the interaction of annual temperature with land use intensity is notably robust, slightly surpassing that between annual temperature and population density. This suggests that in regions with varying land use intensities, temperature significantly impacts vegetation productivity, especially in warmer areas where land use changes exacerbate NPP influences, demonstrated by X5∩X14 (0.5103) > X5∩X12 (0.4948) (Fig 7B, C).

Overall, there exists a marked interactive enhancement effect between natural and anthropogenic factors affecting NPP, characterized by mutual and nonlinear reinforcement. It is evident that no natural factor operates in isolation from others (Fig 7B, C). The interactions among natural factors, such as elevation, precipitation, temperature, and solar radiation, are particularly strong, and the synergy between human activity factors, like population density, GDP density, and land use intensity should also be recognized.

Additionally, ecological detection serves to evaluate the relative significance of natural factors impacting NPP. The results from this detection, shown in Fig 7D, indicate whether statistically significant differences exist among pairs of factors. A marked difference is represented by "Y," while the absence of such a difference is indicated as "N" (Fig 7D). The analysis reveals significant differences among all factors concerning NPP spatial distribution, except between X7 and X5. These notable variations highlight the necessity of considering multiple factors comprehensively to understand the spatial variations in NPP effectively.

**Detection of factor indicative roles.** The risk detector analysed the performance of influencing factors in environments suitable for changes in vegetation NPP and conducted statistical significance tests at a 95% confidence level (Fig 8). Higher NPP values indicate that natural and human activity factors are more favourable to NPP variation. The results show that there are significant differences in the impact of different intervals of explanatory variables on vegetation NPP variation (Fig 8).

Elevation, total solar radiation, distance to roads, distance to cropland, distance to forestland, river network density, and distance to built-up areas all reach their highest NPP values in the first classification interval, with respective values of 812.65 gC m$^{-2}$.a$^{-1}$, 714.64 gC m$^{-2}$.a$^{-1}$, 264.89 gC m$^{-2}$.a$^{-1}$, 399.69 gC m$^{-2}$.a$^{-1}$, 47270491.22 gC m$^{-2}$.a$^{-1}$, and 330.14 gC m$^{-2}$.a$^{-1}$. In contrast, slope, soil moisture, annual mean temperature, annual precipitation, 0°C accumulated temperature, population density, GDP density, land use intensity, and distance to water bodies reach their highest NPP values in the ninth classification interval, with values of 434.19 gC m$^{-2}$.a$^{-1}$, 625.19 gC m$^{-2}$.a$^{-1}$, 860.20 gC m$^{-2}$.a$^{-1}$, 456.42 gC m$^{-2}$.a$^{-1}$, 882.39 gC m$^{-2}$.a$^{-1}$, 429.22 gC m$^{-2}$.a$^{-1}$, 412.99 gC m$^{-2}$.a$^{-1}$, 419.71 gC m$^{-2}$.a$^{-1}$, and 353.06 gC m$^{-2}$.a$^{-1}$, respectively. Evapo-transpiration and distance to grassland reach their highest NPP values in the sixth classification interval, with values of 257.41 gC m$^{-2}$.a$^{-1}$ and 269.66 gC m$^{-2}$.a$^{-1}$, respectively. Aspect reaches its highest NPP value in the fifth classification interval, with a value of 224.63 gC m$^{-2}$.a$^{-1}$.

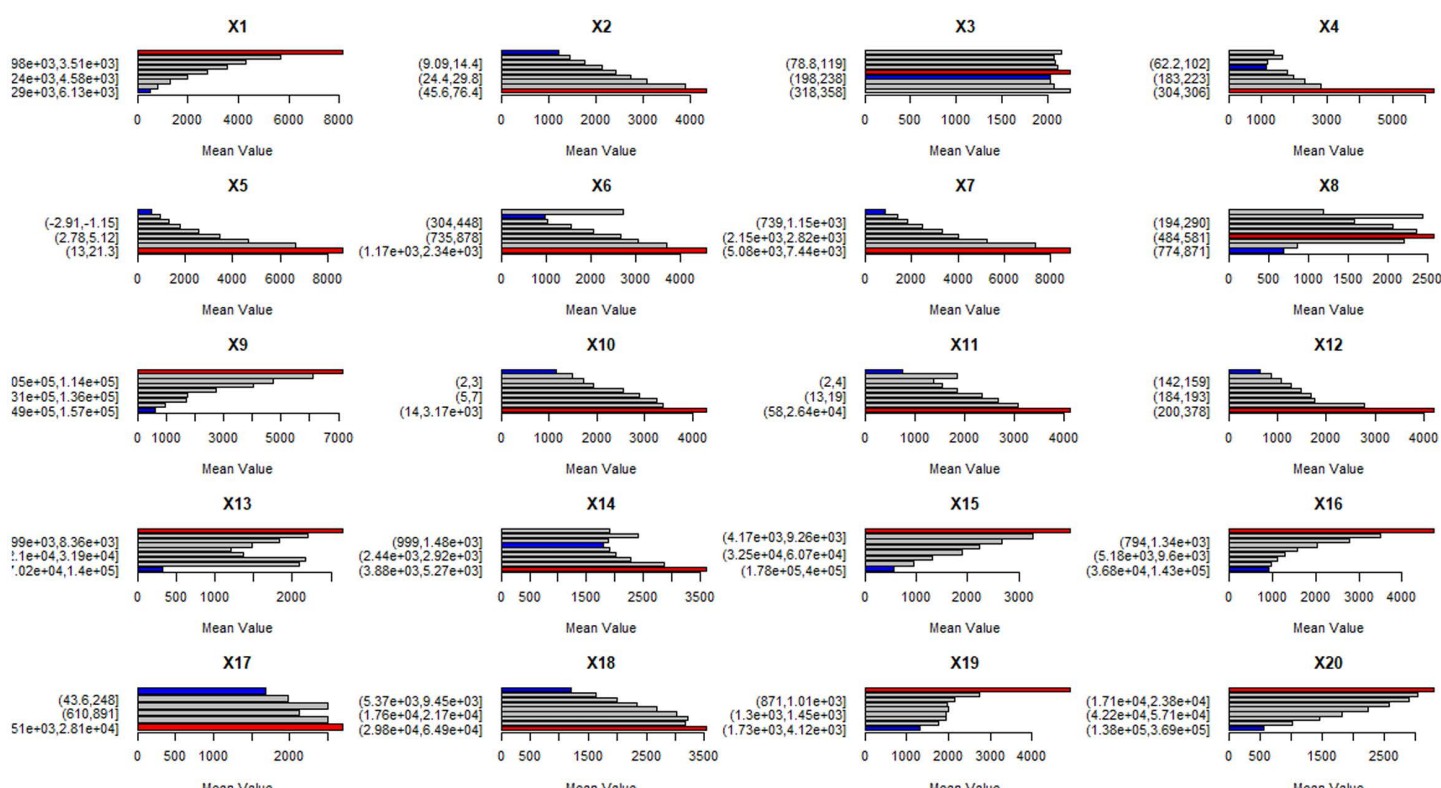

**Fig 8. Risk detection (vegetation NPP variation is classified into three levels: high (red), medium (grey), and low (blue), based on the identified variables).**

## Discussion

### Interactions between natural factors and human activities

This study reveals the combined effects of natural factors and human activities on vegetation NPP, with particular emphasis on the decisive role of their interactions in shaping the spatial variability of NPP. While individual factors, such as altitude, solar radiation, and temperature, significantly influence NPP, the synergistic effects between these factors are more complex and critical. These interactions collectively shape the spatial heterogeneity and changing trends of NPP [25,28].

Firstly, the interaction between altitude and annual precipitation demonstrates that precipitation has a more pronounced impact on vegetation growth in high-altitude regions. At high altitudes, low temperatures limit plant transpiration, while abundant precipitation provides essential water support for vegetation growth. The combined effect of altitude and precipitation on NPP is therefore significantly greater than the influence of either factor alone [64]. Similarly, the interaction between population density and GDP density highlights the amplifying effects of human activities on NPP changes in areas with concentrated population and economic activities. In such regions, intensive land use changes increase resource exploitation, while investments in management and agricultural technology enhance vegetation productivity [65]. Furthermore, the interaction between temperature and human activities is particularly notable. For example, in warm climates, the impacts of agricultural expansion and land development are more apparent, indicating that these factors jointly amplify their promotive or inhibitory effects on NPP [14].

Secondly, the interaction between road distance and forest distance reveals the complex impacts of road construction on vegetation NPP. Although road expansion improves regional economies and transportation accessibility, it may exacerbate deforestation in forested areas, thus negatively affecting vegetation growth. Studies have shown that forests located closer to roads may experience significant declines in NPP due to intensified human activities, whereas forests farther from roads tend to exhibit greater ecological stability [66]. This finding underscores the importance of the spatial relationship between road construction and forest distribution in influencing NPP. Additionally, the interaction between annual average temperature and land-use intensity offers valuable insights. In warmer climates, land-use changes have a particularly pronounced effect on NPP. For instance, in warm regions, agricultural expansion and urbanisation can significantly alter vegetation cover, further amplifying the combined impacts of natural factors and human activities on ecosystems [67].

In summary, the interactions between natural factors and human activities play a pivotal role in NPP changes. The synergistic effects among natural factors such as altitude, precipitation, temperature, and solar radiation contribute to significant spatial heterogeneity, while the interactions between human activity factors such as population density, GDP density, and land-use intensity considerably amplify their impacts on NPP. These interactions not only have profound implications for ecosystem stability and sustainability but also provide a scientific basis for regional ecological management. Therefore, understanding NPP changes requires a shift from analysing the effects of individual factors to a comprehensive exploration of their interactions. Systematic investigations into the combined effects of natural and human factors can offer robust decision-making support for ecosystem conservation and management [54,68].

### Natural factors

**Elevation.** Vegetation NPP varies with elevation, reflecting the ecological differences across the Tibetan Plateau. This study, based on 50 m elevation intervals, used GEE to illustrate NPP changes with altitude (Fig 9). As shown in Fig 9, NPP negatively correlates with elevation; as altitude increases, NPP decreases. This is linked to environmental factors like temperature, oxygen levels, and moisture availability [2].

At 500 m, the mean NPP is 145.017 gC m².a⁻¹, the highest in the study area (Fig 9). As elevation increases, NPP drops significantly to around 83.041 gC m².a⁻¹ at 2000 m, indicating a decline in productivity. Between 1500 and 3000 m, the decline is gradual (Fig 8). Above 3000 m, the decrease accelerates, particularly above 4000 m, where NPP sharply declines with every 500 m increase, falling from 30.169 gC m².a⁻¹ to about 0.34 gC m².a⁻¹, indicating near-zero productivity

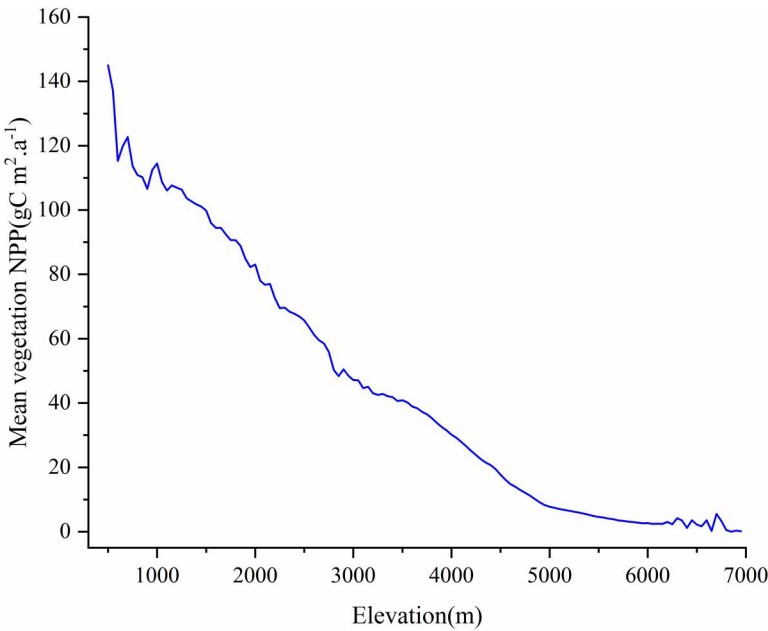

**Fig 9. Relationship between vegetation NPP and elevation on the Tibetan Plateau from 2001 to 2021.**

(Fig 9). At elevations near 7000 m, NPP is almost zero, showing the extreme limitations of high-altitude environments on vegetation growth [69,70].

Therefore, lower-altitude areas benefit from more favorable climatic and soil conditions, supporting vegetation growth, while higher altitudes pose increasing challenges to vegetation productivity.

**Total solar radiation.** The Tibetan Plateau, sensitive to global climate change, is strongly influenced by total solar radiation, which drives photosynthesis and vegetation NPP [71]. Due to its high altitude, low temperatures, and unique monsoon climate, the impact of solar radiation on NPP shows distinct patterns [72,73]. The partial correlation coefficient between NPP and solar radiation ranges from -0.89 to 0.85, indicating significant spatial variation with both positive and negative correlations (Fig 10A).

As shown in Fig 10A, 34.16% of the area shows a positive correlation (partial correlation coefficient > 0). Among these, 7.90% have coefficients between 0.3 and 0.6, 0.36% exceed 0.6, and 25.89% are between 0 and 0.3. Solar radiation notably enhances NPP, particularly in regions with strong radiation and favorable temperature and moisture, like south-western Qinghai, northern Sichuan, and eastern Tibet [70,72–74].

In contrast, 65.84% of the area shows a negative correlation (partial correlation coefficient < 0), where increased solar radiation suppresses NPP. Areas with coefficients less than -0.6, between -0.6 and -0.3, and between -0.3 and 0 represent 2.45%, 24.09%, and 39.30%, respectively. This negative correlation is likely due to high temperatures that reduce photosynthetic efficiency and increase water evaporation, hindering plant growth. These areas are mostly found in northeastern, eastern, and southwestern Qinghai, and parts of Tibet.

**Annual average temperature.** Partial correlation analysis shows that the correlation between vegetation NPP and annual average temperature on the Tibetan Plateau ranges from -0.80 to 0.89, with both positive and negative correlations (Fig 10B). This aligns with studies highlighting temperature's importance for plant growth but showing significant regional variation [75,76], contrasting with conclusions suggesting a general positive correlation without considering spatial heterogeneity [77].

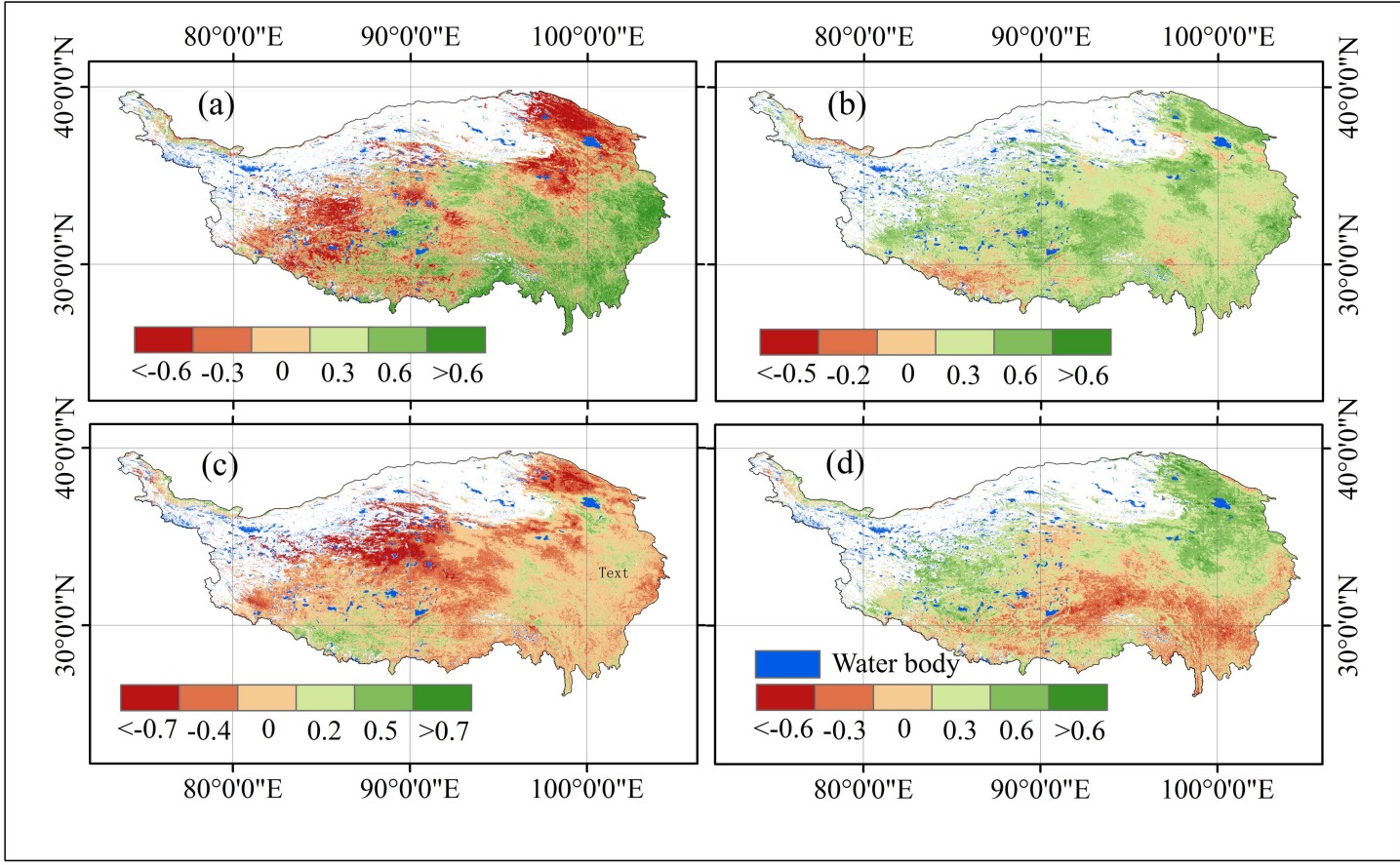

**Fig 10. Partial correlation coefficients between vegetation NPP changes and climatic factors, including total solar radiation (A), annual average temperature (B), accumulated temperature above 0 °C (C), and annual average precipitation (D), on the Tibetan Plateau from 2001 to 2021.**

Positive correlation: As shown in Fig 10B, 31.84% of the area shows a positive correlation (partial correlation coefficient > 0), with 30.02% having coefficients between 0.3 and 0.6 and 1.82% greater than 0.6. Temperature significantly promotes NPP in areas mainly in northeastern Qinghai, Tibet, and Sichuan. However, 52.54% of the area has a weaker effect (0 to 0.3), indicating temperature is not the primary driver of NPP variation in these regions, which are mainly in southern Qinghai, northwestern Sichuan, and Tibet.

Negative correlation: The negative correlation areas (Fig 10B) suggest that rising temperatures suppress NPP. Regions with coefficients below -0.5 account for 0.11%, and between -0.5 and -0.2, 2.98%, indicating high temperatures inhibit growth due to moisture loss. These areas are mainly in southwestern Tibet and northwestern Sichuan. Additionally, 12.53% of the area has a minor negative effect (between -0.2 and 0), suggesting other factors, such as precipitation or solar radiation, may play a larger role in NPP variation [10,78].

**Accumulated temperature above 0 °C.** The partial correlation coefficient between vegetation NPP and accumulated temperature above 0 °C on the Tibetan Plateau ranges from -0.96 to 0.74, showing significant regional variation (Fig 10C).

Positive correlation: Positive correlation areas account for 15.22% of the total area. Of these, regions with coefficients between 0.2 and 0.5, and 0.5 to 0.74, make up 4.23% and 0.16%, respectively. These areas, mainly in the eastern and southeastern Tibetan Plateau and around the Sichuan Basin, show that increased accumulated temperature boosts plant photosynthesis, enhancing NPP under warmer conditions [11,12,79].

Negative correlation: Negative correlation areas cover 84.78% of the total area. Specifically, areas with coefficients below -0.7, between -0.7 and -0.4, and between -0.4 and 0 make up 6.91%, 27.21%, and 50.66%, respectively. In these regions, higher accumulated temperature increases evapotranspiration, reducing plant growth, particularly in the highlands of the Tibetan Plateau[7,80]. While moderate accumulated temperature promotes NPP, excessively high temperatures or water shortages hinder growth [81].

**Annual precipitation.** The relationship between precipitation and NPP on the Tibetan Plateau is complex and region-dependent, with nonlinear characteristics [72]. The partial correlation coefficient between NPP and annual precipitation ranges from -0.89 to 0.90, showing both positive and negative correlations with significant spatial heterogeneity (Fig 10D).

Negative correlation: In negatively correlated areas, regions with coefficients below -0.6, between -0.6 and -0.3, and between -0.3 and 0 make up 1.26%, 15.20%, and 29.33%, respectively. Excessive precipitation may hinder plant growth, with other factors like temperature and radiation playing a more significant role [5,73]. These areas are mainly in eastern and southeastern Tibet, and northwestern Sichuan and Yunnan.

Positive correlation: In positively correlated areas, regions with coefficients between 0.3 and 0.6, and above 0.6 comprise 18.91% and 1.30%, respectively. Increased precipitation enhances NPP, especially in northeastern and southeastern Qinghai, northwestern Tibet, and southwestern Gansu [82].

Overall, while precipitation's impact is limited in most areas, certain regions show stronger correlations, indicating that precipitation can both promote and inhibit vegetation growth.

### Human activity factors

**Distance factors. Distance from forest areas.** Vegetation NPP decreases sharply as the distance from forest areas increases, then stabilises with minor fluctuations in areas farther away (Fig 11A). This suggests a significant positive influence of forest areas on nearby vegetation productivity, which weakens with distance [67]. As shown in Fig 10A, within 20,000 m of forest areas, NPP declines significantly, indicating higher productivity closer to forests due to benefits like improved soil moisture and nutrients, along with reduced human interference. Between 20,000 and 120,000 m, NPP

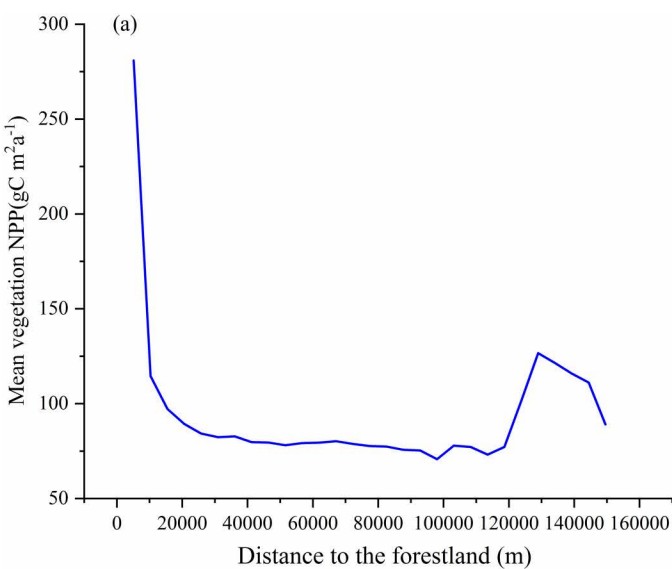
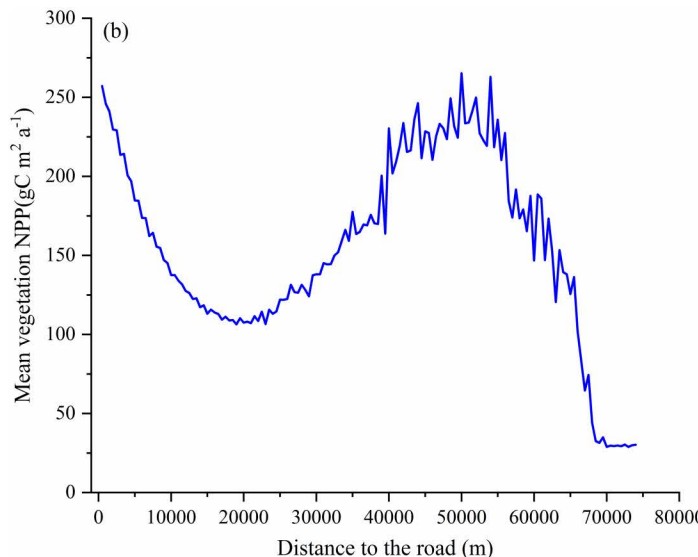

**Fig 11. Variation in vegetation NPP with distance from forest areas (A) and roads (B).**

stabilises at a lower level, showing a reduced forest influence. Beyond 120,000 m, fluctuations in NPP suggest influences from other factors, such as land use, human activities, or ecological variations.

**Distance from roads.** The relationship between NPP and distance from roads follows a trend of initial decline, then a rise, and finally a sharp decrease, reflecting the impact of roads on vegetation (Fig 11B). Within 10,000 m of roads, NPP drops from 250 gC m$^{-2}$·a$^{-1}$ to 100 gC m$^{-2}$·a$^{-1}$, suggesting higher productivity near roads due to factors like accessibility and management. Between 10,000 and 50,000 m, NPP rises to 250 gC m$^{-2}$·a$^{-1}$, likely due to reduced disturbance and ecosystem recovery. Beyond 50,000 m, NPP decreases sharply to below 50 gC m$^{-2}$·a$^{-1}$, stabilising at 30 gC m$^{-2}$·a$^{-1}$ in areas over 70,000 m from roads, where harsh environmental conditions or minimal human activity limit vegetation growth [46].

**Population density.** The correlation between NPP and population density on the Tibetan Plateau is -0.37, indicating a negative relationship. High population density areas near arable land and settlements often face vegetation degradation, soil erosion, and reduced biodiversity due to intensive human activities like grazing and agriculture, leading to lower NPP [83,84]. In contrast, low population density areas, such as nature reserves, show higher NPP due to minimal human disturbance and natural vegetation recovery [85].

Terrain and climate variations also influence this relationship. In high-altitude regions, harsh conditions result in low NPP despite sparse populations. Conversely, regions with moderate elevation and favourable conditions maintain higher NPP through effective management and restoration, even with higher population densities [86]. Some densely populated agricultural clusters achieve higher NPP via intensive practices and vegetation restoration [87].

Rising population density and infrastructure expansion increase resource demand, reducing vegetation cover and ecosystem connectivity, which further lowers NPP [28]. However, recent ecological policies and technological advancements have started reversing NPP declines in some high-density areas [88].

**GDP density.** GDP density on the Tibetan Plateau, reflecting economic output per unit area, is closely linked to urbanisation, land use change, and resource exploitation, which directly and indirectly impact NPP [25,28]. Economic activities are concentrated in cities like Lhasa, Shigatse, and Nyingchi, where high GDP density often corresponds to significant human disturbance and reduced NPP [89].

In high GDP density regions, urban expansion, infrastructure development, and agricultural intensification alter land use, reducing vegetation cover and lowering NPP [25]. Urbanisation often diminishes green spaces, while agricultural expansion can initially raise NPP but may degrade soil and reduce NPP over time. Resource exploitation, such as mining, further damages ecosystems through vegetation destruction, soil erosion, and pollution [30].

Additionally, economic activities increase greenhouse gas emissions, intensifying climate change and indirectly affecting NPP by altering precipitation patterns and temperatures. These changes can exacerbate drought and cause earlier snowmelt, negatively impacting NPP [28].

**Land use change. Land use intensity.** As shown in Fig 12, NPP on the Tibetan Plateau has a non-linear relationship with land use intensity. Moderate human disturbance promotes NPP growth, while excessive disturbance inhibits it [90,91]. This highlights the need for rational land use and vegetation protection to sustain ecosystems [92,93,94].

At low land use intensity (<120), both the mean and standard deviation of NPP remain low and stable. For instance, at an intensity of 100, the mean NPP is 52.29 gC m$^{-2}$·a$^{-1}$, with a standard deviation of 36.59. Moderate land use intensity (130–160) increases both metrics, indicating enhanced vegetation growth but greater variability. For example, at an intensity of 158, the mean NPP is 425.26 gCm$^{-2}$·a$^{-1}$, with a standard deviation of 259.19.

In high-intensity areas (170–200), the mean and variability of NPP rise further. At an intensity of 174, the mean NPP is 381.75 gC m$^{-2}$·a$^{-1}$, with a standard deviation of 229.92. However, extremely high intensities (>200) show diverse trends: some areas see increased NPP with high variability (e.g., at 202, mean NPP is 579.02 gC m$^{-2}$·a$^{-1}$), while others experience declines (e.g., at 238, mean NPP drops to 160.19 gC m$^{-2}$·a$^{-1}$). This suggests that excessive activities, such as urbanisation and industrial expansion, negatively affect vegetation productivity [95].

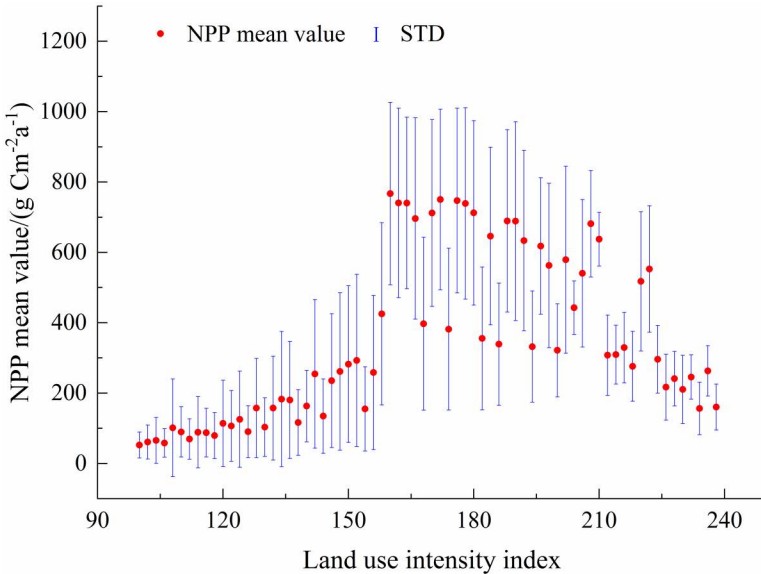

**Fig 12. Relationship between vegetation NPP and land use intensity from 2001 to 2021.**

**Land use conversion.** Land use conversion significantly influences carbon storage and ecosystem services on the Tibetan Plateau. Fig 13A shows major transitions from 2001 to 2021, with frequent exchanges between grassland and unused land. Grassland transformed into unused land, while unused land reverted to grassland and forest. Transitions between cropland and forest were less common, reflecting relative stability, and these changes are driven by human activities, environmental factors, and policies [67,96].

According to Fig 13B, converting 2,705 km² of cropland to forest increased NPP by $1.17\times10^6$ tC, enhancing carbon storage and ecosystem services. Similarly, 4,680 km² of cropland converted to grassland raised NPP by $0.80\times10^6$ tC, while 566 km² converted to water bodies altered NPP by $0.048\times10^6$ tC, benefiting aquatic ecosystems [95]. However, 651 km² of cropland converted to built-up land decreased NPP by $0.095\times10^6$ tC due to urban expansion [67,97].

Conversely, 2,425 km² of forest converted to cropland reduced NPP by $0.87\times10^6$ tC, weakening carbon storage[98]. Similarly, 8,011 km² of grassland converted to cropland caused a significant NPP decline of $2.90\times10^6$ tC, reflecting the adverse effects of agricultural expansion[99]. Other notable changes include 337 km² of water bodies converted to cropland, altering NPP by $0.12\times10^6$ tC, and 362 km² of unused land converted to cropland, increasing NPP by $0.13\times10^6$ tC, expanding agricultural areas [100,101].

**Grazing.** Grazing has a dual impact on vegetation NPP, with both benefits and drawbacks [92]. Moderate grazing reduces plant competition, enhances photosynthesis, improves soil fertility, and stabilizes grassland ecosystems [94]. In contrast, overgrazing causes vegetation degradation, soil damage, and NPP decline, replacing high-productivity plants with low-productivity species, and exacerbating soil compaction and erosion [91,93]. Climate change further interacts with grazing, affecting NPP by altering temperatures and precipitation. While warming may extend the growing season, excessive grazing can offset these benefits by suppressing plant growth [95].

**Other factors.** While individual factors such as proximity, river and road density, slope, aspect, soil moisture, and moderate evapotranspiration have limited explanatory power alone, their combined effects significantly influence vegetation NPP changes. NPP variation on the Tibetan Plateau results from complex interactions between natural and human factors.

Proximity to forests, grasslands, and water bodies generally corresponds to higher NPP due to favorable moisture and vegetation conditions, whereas proximity to arable land and built-up areas often indicates lower NPP due to human

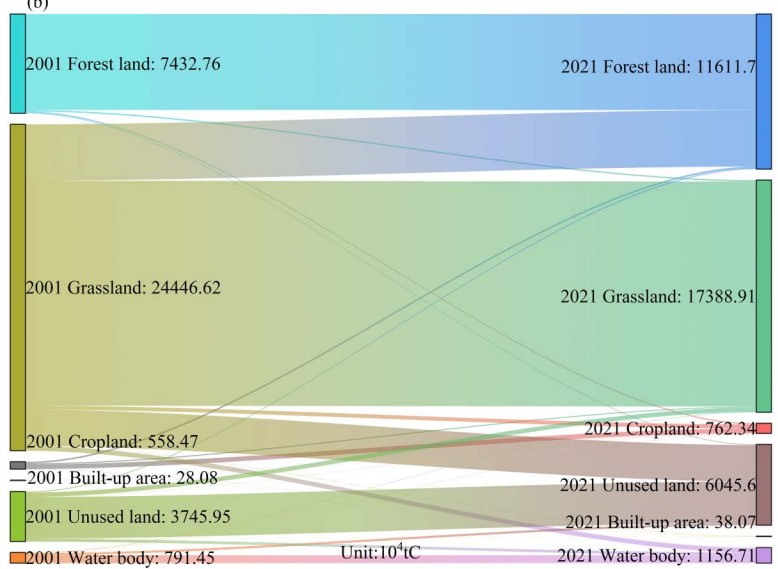

**Fig 13. Land use transitions (A) and corresponding vegetation NPP changes (B) on the Tibetan Plateau from 2001 to 2021.** Note: 2001 and 2021 in the figure represent the years 2001 and 2021, respectively.

disturbances [93]. Dense road and river networks fragment ecosystems, reducing NPP [46]. Slope and aspect impact vegetation by affecting soil erosion and solar radiation, with south-facing slopes usually showing higher NPP. Moderate soil moisture and evapotranspiration promote vegetation growth, but excessive evapotranspiration may inhibit NPP.

## Comparison with other studies

Our study has notable methodological advantages, including in-depth exploration of driving factor interactions, regional subdivision analysis, and the integration of multi-source data. These strengths provide reliable insights into the driving

mechanisms of vegetation NPP changes on the Tibetan Plateau, supporting ecological management and sustainable development.

**Similarities.** Consistent with previous studies, we focus on the Tibetan Plateau, analyzing the spatial and temporal variations in vegetation NPP. Similar to Xu et al. [69] and Piao et al. [2], we identify fluctuating upward trends in NPP and regional differences, with higher values in the southeast and lower values in the northwest cold desert, as noted by Zhang et al. [20]. Key drivers such as solar radiation, temperature, evapotranspiration, soil conditions, land use changes, and population density are confirmed in our research, aligning with studies like Sun et al. [102].

**Differences.** Unlike prior studies, we applied advanced methods like Sen's slope analysis, Hurst index, and Mann-Kendall test, combined with the OPGD model, to reveal non-linear relationships and spatial heterogeneity [103]. This enables us to quantify the interactions between natural factors and human activities, offering new perspectives often overlooked in other research [104,105].

Our regional subdivision analysis delves deeper into northeastern, southeastern, northwestern, and southwestern areas, uncovering spatial heterogeneity and providing new insights for targeted ecological protection strategies [106]. By integrating multi-source remote sensing and environmental data on the GEE platform, we achieved large-scale, long-term analyses with greater precision, enhancing the reliability and applicability of our findings [107,108].

### Research limitations

**Data limitations.** While we integrated multi-source remote sensing and environmental data, spatial and temporal resolution constraints may have missed subtle ecological changes [107,109]. Inaccuracies in key environmental variables could also affect the results [110]. Additionally, the long-term lag effects of ecosystem changes mean the trends observed may not fully capture future ecological evolutions [59].

**Model assumptions.** The methods used, including Sen's trend analysis and the OPGD model, are effective for identifying trends and drivers. However, assumptions and parameter choices could impact the results' precision, especially when handling non-linear and complex interactions [103–105].

**Complexity of driving factors.** While various natural and human-driven factors affecting NPP were identified, the interactions between them remain complex and may not be fully captured [106,108]. Socio-economic and policy impacts on NPP were not thoroughly evaluated due to the lack of detailed social data [2,69].

**Regional heterogeneity.** Although regional subdivision analysis was conducted, small-scale heterogeneity and microclimatic effects may not have been fully captured, particularly in alpine desert and mountainous areas where terrain and micro-environments significantly affect NPP [15,21,67].

In conclusion, these limitations highlight the need for higher-resolution data, optimized models, and a focus on socio-economic factors in future research to better understand NPP changes on the Tibetan Plateau and improve ecosystem management [95,107,108].

### Conclusion

Our study not only reveals the spatiotemporal patterns and key driving factors behind the changes in Net Primary Productivity (NPP) on the Tibetan Plateau but also delves into the interactions between natural and anthropogenic factors using non-parametric trend methods and innovative spatial statistical models. This research provides crucial scientific evidence for the ecological conservation and sustainable development of the Tibetan Plateau.

Firstly, the study finds that the overall NPP of vegetation on the Tibetan Plateau exhibits nonlinear fluctuations, with significant spatial variability. Multi-regional analysis shows a marked increase in NPP in the northeastern and southeastern regions, while the alpine desert areas in the northwest and southwest maintain low NPP levels. These regional differences are closely linked to topography, climatic conditions, and the intensity of human activities.

Further trend analysis indicates that over 55% of the region shows an increasing trend in NPP, primarily concentrated in the northern, western, and southern peripheral areas. In contrast, NPP remains stable ($p \geq 0.05$) in 34% of the area, while over 10% shows a decline in NPP, particularly in the eastern and southern regions with frequent human activity. Areas exhibiting weak persistence in NPP changes account for 40.63%, indicating that some ecosystems display a degree of resilience under the dual pressures of climate change and human activities.

The study also identifies multiple driving factors behind NPP changes. Elevation and climatic factors (such as solar radiation, mean annual temperature, and accumulated temperature above 0°C) have significant impacts on NPP variation, with climatic factors showing both positive and negative correlations, revealing notable spatial heterogeneity. Additionally, human activity factors such as forest proximity, road density, population density, and GDP density also play important roles in NPP changes. These natural and anthropogenic factors exhibit spatial heterogeneity and jointly intensify NPP fluctuations.

Through the use of the optimised Geographical Detector (OPGD) model, we conducted an in-depth analysis of the synergistic effects between natural and human factors. The findings demonstrate significant nonlinear and bilinear enhancement effects between natural and anthropogenic factors, reflecting the regulatory influence of these complex interactions on NPP changes. This suggests that the driving mechanisms behind NPP variation cannot be fully captured through simple linear analyses; instead, the intricate relationships among multiple factors must be considered.

Lastly, by integrating multi-source remote sensing and environmental data and utilising the Google Earth Engine (GEE) platform, we conducted large-scale, long-term NPP analyses on the Tibetan Plateau. This approach not only enhances the reliability of the research outcomes but also provides vital scientific support for the long-term monitoring and management of the Tibetan Plateau's ecosystem, with broad applicability.

## Supporting information

**S1 Appendix. GEE code.**
(DOCX)

**S2 Appendix. MATLAB code.**
(DOCX)

## Author contributions

**Data curation:** Jie Tang, Xinghong Peng.

**Methodology:** Jie Tang, Xinghong Peng, Wenfu Peng.

**Resources:** Xinghong Peng.

**Software:** Wenfu Peng.

**Writing – original draft:** Jie Tang, Xinghong Peng.

**Writing – review & editing:** Wenfu Peng.

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
