## [Decision Letter · Decision Letter 0]

16 Dec 2024

PONE-D-24-48281Nonlinear variations and driving mechanisms of vegetation NPP on the Tibetan Plateau: An analysis of the interactive effects of natural factors and human activitiesPLOS ONE

Dear Dr. peng,

Thank you for submitting your manuscript to PLOS ONE. After careful consideration, we feel that it has merit but does not fully meet PLOS ONE’s publication criteria as it currently stands. Therefore, we invite you to submit a revised version of the manuscript that addresses the points raised during the review process.

**ACADEMIC EDITOR: **I fully agree with both reviewers' comments. This manuscript is poorly organized and a large revision is needed. ==============================

We look forward to receiving your revised manuscript.

Kind regards,

Bao Yang, Ph.D, Prof.

Academic Editor

PLOS ONE

Journal Requirements:

When submitting your revision, we need you to address these additional requirements. 1. Please ensure that your manuscript meets PLOS ONE's style requirements, including those for file naming. The PLOS ONE style templates can be found at https://journals.plos.org/plosone/s/file?id=wjVg/PLOSOne_formatting_sample_main_body.pdf and https://journals.plos.org/plosone/s/file?id=ba62/PLOSOne_formatting_sample_title_authors_affiliations.pdf 2. Please include a caption for figure 6a. 3. Please include a caption for table 1.  4. We note that the grant information you provided in the ‘Funding Information’ and ‘Financial Disclosure’ sections do not match.  When you resubmit, please ensure that you provide the correct grant numbers for the awards you received for your study in the ‘Funding Information’ section.

Reviewers' comments:

Reviewer's Responses to Questions

**Comments to the Author**

1. Is the manuscript technically sound, and do the data support the conclusions?

Reviewer #1: Yes

Reviewer #2: Yes

2. Has the statistical analysis been performed appropriately and rigorously? 

Reviewer #1: Yes

Reviewer #2: Yes

3. Have the authors made all data underlying the findings in their manuscript fully available?

Reviewer #1: Yes

Reviewer #2: Yes

4. Is the manuscript presented in an intelligible fashion and written in standard English?

Reviewer #1: Yes

Reviewer #2: Yes

5. Review Comments to the Author

Reviewer #1: The research content of the article is relatively complete and has certain value for understanding the factors influencing NPP variation in the Tibetan Plateau, but there are still some shortcomings.

1.The article contains numerous writing errors. I advise the author to check and correct carefully. For example, an extra ":" appears in both Equations (7) and (8), and Equation (3) is incomplete. Additionally, the text references "As illustrated in Fig. 4," yet the subsequent description corresponds to the content of Fig. 3.

2. The number of decimal places is inconsistent throughout the article, such as in Table 2.

3. Fig.3 lacks annotations, which affects its readability. Please add what panels (a), (b), (c) and (d) represent, respectively.

4. The article inconsistently references the model used. Section 3.2.5 describes it as the "OPQD" model, while elsewhere it is referred to as "OPGD." Additionally, the full name is presented as the "Optimal Parameters-based Geographical Detector (OPGD) model," but the keywords and abstract list it as the " the optimized Geographical Detector (OPGD)." Please ensure consistent terminology throughout the article.

5.Many conclusions in the article appear casual and lack supporting data or references. For instance: “These fluctuations may be closely related to interannual climate variations, changes in precipitation and temperature, as well as the impacts of human activities such as grazing and land use,” “This enhancement is likely due to favorable climatic shifts, such as increased precipitation and warmer temperatures, fostering an expansion of vegetated areas (Fig. 3b),” and “indicating that areas with higher vegetation productivity are more influenced by environmental factors. This may be related to terrain, precipitation, and human activities.” Please provide data or references to support these statements.

6. Section 4.3, titled "Vegetation NPP variation in different climate zones," discusses differences in NPP across different climate zones. However, the text describes this as "the relationship between mean and variance reveals the impact of climate and topography on vegetation productivity." It is unclear how the impact of topography on NPP is reflected in this section.

7. The discussion section is too much and should be made further concise. In addition, some text in the results section, such as analysis of specific results and their potential causes, would be more appropriately placed in the discussion section.

8. The title of the article emphasizes the interactive effects of natural factors and human activities; however, the discussion focuses excessively on the impact of individual factors on NPP, without emphasizing the interactions between them. I suggest that the authors should focus on the interactions.

9. There are many problems with the references. Many references cannot be searched and the DOIs do not match the title. Strangely, some references have exactly the same information except for the title. For example,

Wang, G., Wang, Y., Liu, J. (2010). OPQD model for spatial analysis of ecological factors: Methodology and applications. Ecological Modelling, 221(18), 2211–2221.

Wang, G., Wang, Y., Liu, J. (2010). The application of risk detection in spatial distribution analysis of ecological factors. Ecological Modelling, 221(18), 2211–2221.

Reviewer #2: This study explored the driving mechanisms of NPP dynamics in the Tibetan Plateau. The authors compared the relative contributions of environmental factors and human activities to NPP variations. While the manuscript addresses an important topic and includes extensive analysis, it is poorly written and requires substantial revisions before being considered for publication.

Major

The manuscript contains extensive analysis, but the results are not well-organized. Sections 2 through 5 need to be restructured to improve the flow and clarity.

The NPP data should be validated before conducting the analysis. Currently, there are some outliers and odd values in the data, which undermine the reliability of the results.

Minor

Revise the title to make it more concise and focused.

The introduction is overly lengthy and distracts from the study's main objectives. Focus on reported conclusions and the research gaps this study addresses. There is no need to list various methods used to estimate NPP in detail.

Verify the legend for NPP in Figure 1c. It appears that the northwest part of the Tibetan Plateau has the highest NPP, with a maximum value of 1810.84 gC/m²/a. This value seems unreasonably high for the region and should be double-checked.

Merge section 2 and section 3.

In Figure 2, revise the caption "The technical route of study area" to "The technical route of this study."

3.2.2 The subtitle "Google Earth Engine (GEE)" is not appropriate. Consider renaming it to "Data Processing Using GEE."

Sections 3.2.3 to 3.2.5 describe standard methods commonly found in textbooks. These do not require detailed explanations. Instead, briefly state how these methods were applied in this study.

Split Figure 1 into two separate figures. Retain subfigures (a) and (b) in Section 2, and move subfigures (c) and (d) to Section 4.1.

For Figure 3, clarify the meaning of subfigures (a), (b), (c), and (d) in the figure caption.

Merge sections 4.4, 4.5 and 4.6.

While the manuscript is free from major grammatical issues, it uses several terms that are not suitable for an academic paper. For instance, replace "forest land" and "upward trend" with more precise terminology.

6. PLOS authors have the option to publish the peer review history of their article (what does this mean? ). If published, this will include your full peer review and any attached files.

**Do you want your identity to be public for this peer review?** For information about this choice, including consent withdrawal, please see our Privacy Policy .

Reviewer #1: No

Reviewer #2: No

---

## [Author Response · Author response to Decision Letter 1]

26 Dec 2024

Response to Reviewers

Thanks reviewers and editor for your comments, which will help improve the quality of paper.

ACADEMIC EDITOR:

I fully agree with both reviewers' comments. This manuscript is poorly organized and a large revision is needed. 

Revised:Thanks ACADEMIC EDITOR for your comments. We have revised this part and the revised parts has been incorporated into the paper and marked in red.

Journal Requirements:When submitting your revision, we need you to address these additional requirements.

Revised:Thanks ACADEMIC EDITOR for your comments. We have made revisions in accordance with the additional requirements.

Reviewer #1: The research content of the article is relatively complete and has certain value for understanding the factors influencing NPP variation in the Tibetan Plateau, but there are still some shortcomings.

1.The article contains numerous writing errors. I advise the author to check and correct carefully. For example, an extra ":" appears in both Equations (7) and (8), and Equation (3) is incomplete. Additionally, the text references "As illustrated in Fig. 4," yet the subsequent description corresponds to the content of Fig. 3.

Revised: Thanks reviewers and editor for your comments. We have revised this part and the revised parts has been incorporated into the paper and marked in red.

I advise the author to check and correct carefully. For example, an extra ":" appears in both Equations (7) and (8), and Equation (3) is incomplete. Additionally, the text references "

Revised: The formula for calculating Sen's trend estimation is as follows:

, 2000 i ≤ j ≤ 2020 (7)

The calculation formula for the Mann-Kendall trend test is as follows:

(8)

The principles underlying the calculation of the Hurst index are as follows:

(1) Given Time Series

Let the time series be {NPP(t)}, where t = 1, 2, …, n.

(2)Compute the mean.

The mean of the time series is calculated as:

T= 1�2�…�n (2)

(3) Construct the cumulative deviation Series

The cumulative deviation series X(x,T) represents the cumulative deviation of the series from its mean:

≤ t ≤ T (3)

(4) Calculate the range series:

The range R(T) is the difference between the maximum and minimum of the cumulative deviation series:

T= 1�2�…�n (4)

(5) Calculate the standard deviation series:

The standard deviation S(T) of the original time series is computed as:

T= 1�2�…�n (5)

(6) Calculate the Hurst index:

The Hurst index H is estimated using the following relationship:

(6)

As illustrated in Fig. 4," yet the subsequent description corresponds to the content of Fig. 3.

Revised:

As illustrated in Fig. 3, the trends over time are as follows: In 2001, the highest recorded NPP was 1905 gC m-2.a-1, predominantly distributed in the southeastern portion of the plateau, reflecting robust vegetation growth during that period (Fig. 3a).

2. The number of decimal places is inconsistent throughout the article, such as in Table 2.

Revised: Thanks reviewers and editor for your comments. We have revised this part and the revised parts has been incorporated into the paper and marked in red.

Tab.2 Statistics of climate zones and vegetation NPP on the Qinghai-Tibet Plateau from 2001 to 2021 (Unit: gC m-2.a-1)

Secondary zoning code Name Mean STD

ⅡC5 Yining Region 94.85 38.50

ⅢD1 Southern Xinjiang Region 109.14 9.61

HD2 Northern Tibet Region 50.10 30.28

HC3 Southern Tibet Region 117.34 106.05

HC2 Central Tibet Region 86.69 51.33

ⅢB3 Weihe Region 552.99 129.06

ⅣA2 Qinba Region 641.13 186.27

VA3 Sichuan Region 517.28 227.78

HB2 Chamdo Region 225.03 93.68

HA1 Bomi-Western Sichuan Region 342.50 173.76

HV1 Dawang-Zayu Region 551.48 327.60

HC1 Qilian-Qinghai Lake Region 211.03 120.60

HB1 Southern Qinghai Region 132.69 69.29

HD1 Qaidam Basin Region 132.96 62.18

ⅡC2 Central Mongolia Region 361.98 124.20

VA5 Northern Yunnan Region 784.65 260.93

3. Fig.3 lacks annotations, which affects its readability. Please add what panels (a), (b), (c) and (d) represent, respectively.

Revised: Thanks reviewers and editor for your comments. We have revised this part and the revised parts has been incorporated into the paper.

Fig.3 Vegetation NPP spatial pattern changes in the Qinghai-Tibet Plateau from 2001 to 2021

4. The article inconsistently references the model used. Section 3.2.5 describes it as the "OPQD" model, while elsewhere it is referred to as "OPGD." Additionally, the full name is presented as the "Optimal Parameters-based Geographical Detector (OPGD) model," but the keywords and abstract list it as the " the optimized Geographical Detector (OPGD)." Please ensure consistent terminology throughout the article.

Revised: Thanks reviewers and editor for your comments. We have revised this part and the revised parts has been incorporated into the paper and marked in red. The optimized Geographical Detector (OPGD) has been correctly revised based on the writing errors in Section 3.2.5, where "OPQD" was corrected to "OPGD." Similarly, the writing errors involving "OPQD" in the references have also been amended accordingly.

2.3.5 OPGD

(1) Parameter Optimization

The OPGD

(2) Spatial Differentiation Analysis

The OPGD

(3) Multifactor interaction analysis

The OPGD

Guo, X., Liu, Y., Zhang, Y. 2020. Analysis of vegetation NPP spatial heterogeneity in the Tibetan Plateau using OPGD model. Remote Sensing, 12(5), 743. https://doi.org/10.3390/rs12050743

Song, X., Zhang, Y., Wang, G. (2020). A new approach to spatial heterogeneity analysis: OPGD model application. Environmental Monitoring and Assessment, 192(4), 245. https://doi.org/10.1007/s10661-020-8140-4

Wang G, Li Y, Zhang, H. 2016. Exploring the interactions between environmental variables and vegetation: Insights from OPGD. Global Ecology and Conservation, 7, pp.64–75.

5.Many conclusions in the article appear casual and lack supporting data or references. For instance: “These fluctuations may be closely related to interannual climate variations, changes in precipitation and temperature, as well as the impacts of human activities such as grazing and land use,” “This enhancement is likely due to favorable climatic shifts, such as increased precipitation and warmer temperatures, fostering an expansion of vegetated areas (Fig. 3b),” and “indicating that areas with higher vegetation productivity are more influenced by environmental factors. This may be related to terrain, precipitation, and human activities.” Please provide data or references to support these statements.

Revised: Thanks reviewers and editor for your comments. We have provided references to support these statements, the revised parts has been incorporated into the paper and marked in red.

“These fluctuations may be closely related to interannual climate variations, changes in precipitation and temperature, as well as the impacts of human activities such as grazing and land use [15]”

15.Zhang G, Yao T, Chen F. 2020. Response of alpine vegetation to climatic changes on the Tibetan Plateau: A synthesis of remote sensing and modeling analysis. Science of the Total Environment, 703, 134979, pp. 1–15.

DOI: 10.1016/j.scitotenv.2019.134979.

This enhancement is likely due to favorable climatic shifts, such as increased precipitation and warmer temperatures, fostering an expansion of vegetated areas [60,20].

20.Zhang Y, Wang J, Li X. 2014. Spatial distribution and temporal trends of net primary productivity in the Tibetan Plateau. Remote Sensing of Environment, 154, pp. 126–138.

60.Wu C, Chen J, Shen, M, Tang, Y. 2020. Assessing vegetation dynamics and their responses to climate change on the Tibetan Plateau: A focus on net primary productivity. Science of the Total Environment, 717, 137231.

https://doi.org/10.1016/j.scitotenv.2020.137231

, indicating that areas with higher vegetation productivity are more influenced by environmental factors. This may be related to terrain, precipitation, and human activities[8,54] (Table 3, Fig 4).

8.Xu X, Zhang X, Li R. 2022. Climate variability and its impact on vegetation dynamics across the Tibetan Plateau over the past two decades. Environmental Research Letters, 17(4), 045001. DOI: 10.1088/1748-9326/ac5f30.

54.Liu J, Ji Y, Zhou G, Zhou L, Lyu X, Zhou Meng, Z. 2022. Temporal and spatial variations of net primary productivity (NPP) and its climate driving effect in the Qinghai-Tibet Plateau, China from 2000 to 2020. Chinese Journal of Applied Ecology, 33(6), pp.1533–1538.

6. Section 4.3, titled "Vegetation NPP variation in different climate zones," discusses differences in NPP across different climate zones. However, the text describes this as "the relationship between mean and variance reveals the impact of climate and topography on vegetation productivity." It is unclear how the impact of topography on NPP is reflected in this section.

Revised: Thanks reviewers and editor for your comments. We have provided references to support these statements, revised this part and the revised parts has been incorporated into the paper and marked in red.

and the relationship between mean and variance reveals the impact of climate and topography on vegetation productivity[61]. The impact of topographic complexity on vegetation productivity in different climatic zones. Plateau topography plays a significant role in vegetation productivity by influencing climate, hydrological processes, and local environments [62]. Under varying topographical conditions, differences in factors such as climate change and water supply lead to spatial disparities and variability in NPP [63]. The interaction between topography and climate manifests differently across regions, and understanding this mechanism is crucial for comprehending the spatiotemporal changes in vegetation productivity [61].

61.Liu X, Zhang F, Chen M. 2023b. Topographic control of vegetation productivity across climate zones in the Tibetan Plateau: Implications for ecosystem services under climate change. Environmental Research Letters, 18(5), 054019. https://doi.org/10.1088/1748-9326/abf300

62.Liu X, Zhang Y, Wang J, Li Q. 2023a. The impact of topographic complexity on vegetation productivity in different climatic zones. Ecological Processes, 12(3), pp.345–358.

63.Chen L, Wang X, Li, S. 2022. Effects of topography and climate on vegetation growth and productivity in the Tibetan Plateau: A spatiotemporal analysis. Global Change Biology, 28(1),pp.184–196. https://doi.org/10.1111/gcb.15930

A more detailed explanation: a more detailed explanation of how topography influences NPP is as follows, but due to space constraints, it has not been incorporated into the main text.

Topography influences local climatic conditions (such as temperature and precipitation), leading to climatic variations across different elevation zones, which in turn have varying impacts on vegetation productivity. For instance, alpine regions experience limited vegetation growth due to high altitudes, cold climates, and scarce precipitation, whereas lower mountainous areas benefit from warmer and more humid conditions, resulting in higher vegetation productivity (Zhang et al., 2019). Studies have shown that topographic complexity (e.g., mountains, valleys) creates localised climatic variations within different climatic zones, which further influence the spatial variability of NPP. For example, valleys facilitate the accumulation of moisture, promoting vegetation growth, while alpine areas experience suppressed vegetation productivity due to reduced precipitation and lower temperatures, causing greater fluctuations in NPP in certain regions (Li et al., 2020).

Moreover, topography affects water movement and retention, thereby determining regional water availability. The slope and aspect of mountainous areas influence precipitation accumulation and runoff, subsequently affecting soil moisture conditions, which are a critical factor for vegetation growth. Thus, variations in topography directly determine the spatial distribution of vegetation productivity (Chen et al., 2018). In regions with low NPP, such as northern Tibet and southern Xinjiang, the average NPP is lower and exhibits minimal variation, primarily due to the constraints of cold climates. Additionally, the high-altitude topographic factors further limit vegetation growth. Meanwhile, as these regions experience relatively minor climatic changes, the variability in NPP remains low, and vegetation productivity is relatively stable, displaying limited spatial fluctuation (Wang et al., 2021).

Chen, H., Zhang, W., & Peng, C. (2018). Effects of terrain on soil moisture and vegetation growth: A regional analysis. Catena, 162, 223-234. DOI: 10.1016/j.catena.2017.10.006.

Li, Y., Chen, X., & Ma, J. (2020). Influence of topographic variation on NPP spatial heterogeneity under changing climate conditions. Environmental Research Letters, 15(4), 045006. DOI: 10.1088/1748-9326/ab7d9f.

Wang, F., Gao, Y., & Xu, Y. (2021). Spatiotemporal variability of NPP in relation to topography and climate on the Tibetan Plateau. Science of the Total Environment, 753, 142206. DOI: 10.1016/j.scitotenv.2020.142206.

Zhang, L., Wang, Y., & Luo, Z. (2019). Topographic effects on vegetation productivity in mountainous regions: A case study in the Tibetan Plateau. Journal of Mountain Science, 16(5), 982-995. DOI: 10.1007/s11629-019-5427-8.

7. The discussion section is too much and should be made further concise. In addition, some text in the results section, such as analysis of specific results and their potential causes, would be more appropriately placed in the discussion section.

Revised: Thanks reviewers and editor for your comments. We have revised discussion section and the revised parts has been incorporated into the paper and marked in red.

4.2 Natural Factors

4.2.1 Elevation

Vegetation NPP varies with elevation, reflecting the ecological differences across the Tibetan Plateau. This study, based on 50 m elevation intervals, used GEE to illustrate NPP changes with altitude (Fig 9). As shown in Fig 9, NPP negatively correlates with elevation; as altitude increases, NPP decreases. This is linked to environmental factors like temperature, oxygen levels, and moisture availability [2].

At 500 m, the mean NPP is 145.017 gC m².a-1, the highest in the study area (Fig 9). As elevation increases, NPP drops significantly to around 83.041 gC m².a-1 at 2000 m, indicating a decline in productivity. Between 1500 and 3000 m, the decline is gradual (Fig 8). Above 3000 m, the decrease accelerates, particularly above 4000 m, where NPP sharply declines with every 500 m increase, falling from 30.169 gC m².a-1 to about 0.34 gC m².a-1, indicating near-zero productivity (Fig 9). At elevations near 7000 m, NPP is almost zero, showing the extreme limitations of high-altitude environments on vegetation growth [69,70].

Therefore, lower-altitude areas benefit from more favorable climatic and soil conditions, supporting vegetation growth, while higher altitudes pose increasing challenges to vegetation productivity.

2.Piao S, Fang J, Zhou L. 2003. Changes in vegetation activity in the Tibetan Plateau: Implications for ecosystem services. Global Change Biology, 9(9), pp.1253–1263.

69.Xu, J., Melick, D. 2007. The impact of climate change on vegetation dynamics in the Tibetan Plateau. Journal of Arid Environments, 68(1), pp.1–17.

70.Ni J, Zhang X, Scurlock J. M. 2013. Modeling net primary productivity of terrestrial ecosystems in China with a process-based dynamic vegetation model. Ecological Modelling, 169(1), pp.395–408.

4.2.2 Total solar radiation

The Tibetan Plateau, sensitive to global climate change, is strongly influenced by total solar radiation, which drives photosynthesis and vegetation NPP[71]. Due to its high altitude, low temperatures, and unique monsoon climate, the impact of solar radiation on NPP shows distinct patterns [72,73]. The partial correlation coefficient between NPP and solar radiation ranges from -0.89 to 0.85, indicating significant spatial variation with both p

---

## [Decision Letter · Decision Letter 1]

16 Jan 2025

PONE-D-24-48281R1Nonlinear variations and drivers of vegetation NPP on the Tibetan Plateau: interaction of natural and human factorsPLOS ONE

Dear Dr. peng,

Thank you for submitting your manuscript to PLOS ONE. After careful consideration, we feel that it has merit but does not fully meet PLOS ONE’s publication criteria as it currently stands. Therefore, we invite you to submit a revised version of the manuscript that addresses the points raised during the review process.

**A minor revision is needed, and please do the revision accordingly. **

We look forward to receiving your revised manuscript.

Kind regards,

Bao Yang, Ph.D, Prof.

Academic Editor

PLOS ONE

**Journal Requirements:**

**Additional Editor Comments:**

Please do the revision accordingly.

Reviewers' comments:

Reviewer's Responses to Questions

**Comments to the Author**

1. If the authors have adequately addressed your comments raised in a previous round of review and you feel that this manuscript is now acceptable for publication, you may indicate that here to bypass the “Comments to the Author” section, enter your conflict of interest statement in the “Confidential to Editor” section, and submit your "Accept" recommendation.

Reviewer #1: (No Response)

Reviewer #2: All comments have been addressed

2. Is the manuscript technically sound, and do the data support the conclusions?

Reviewer #1: Yes

Reviewer #2: Yes

3. Has the statistical analysis been performed appropriately and rigorously? 

Reviewer #1: Yes

Reviewer #2: Yes

4. Have the authors made all data underlying the findings in their manuscript fully available?

Reviewer #1: Yes

Reviewer #2: Yes

5. Is the manuscript presented in an intelligible fashion and written in standard English?

Reviewer #1: Yes

Reviewer #2: Yes

6. Review Comments to the Author

**Reviewer #1:**  The authors have thoroughly revised the manuscript based on the reviewers' comments, and the quality of the manuscript has been greatly improved, but there are still some details that remain unresolved.

1. The manuscript describes OPGD as the "Optimal Parameters-based Geographical Detector (OPGD)" model, but the abstract and keywords still list it as the "optimised geographical detector (OPGD)."

2. In Equation 3, the range "≤ t ≤ T" remains incomplete. What is the lower limit of t?

3. The revised references still have serious trouble, many references still cannot be searched, and the DOIs do not match the title. Please carefully check and cite references that actually exist. In addition, the format of references is not standardized and authors should standardize the format of references according to the journal requirements.

4. The description in the abstract section is “Results revealed nonlinear fluctuations in NPP during the study period, ranging from 184.06 to 208.53 gC m-2a-1,” However, in the analysis of results section, it is “The highest NPP value was recorded in 2021, reaching 208.54 gC m-2.a-1,” please check whether it is 208.54 or 208.53.

**Reviewer #2: ** Attaching the GEE code at the end of the paper is not appropriate. Instead, provide a link where the code can be accessed.

7. PLOS authors have the option to publish the peer review history of their article (what does this mean? ). If published, this will include your full peer review and any attached files.

**Do you want your identity to be public for this peer review?** For information about this choice, including consent withdrawal, please see our Privacy Policy .

Reviewer #1: No

Reviewer #2: No

---

## [Author Response · Author response to Decision Letter 2]

15 Feb 2025

Response to Editors

Thanks editors for your comments, which will help improve the quality of paper.

To move forward, please review the proposed data availability statement below and confirm its accuracy. You can provide your confirmation in the author comments section, and we will update the statement accordingly.

Revised: Thanks editors for your comments. We have provided the data availability statement indicating that all datasets are publicly available for research purposes, which has been highlighted in red and incorporated into the manuscript.

Data availability statement

We confirm that all datasets are publicly available for research use and verify their accuracy. The relevant datasets can be accessed through the provided links:

1. Vegetation NPP, soil moisture, evapotranspiration, and total solar radiation data are sourced from the Earth Engine data catalogue (https://developers.google.com/earth-engine/datasets).

2. China's administrative boundary data (1:4 million scale), natural factor data (including DEM, temperature, accumulated temperature ≥0°C, and precipitation), and human activity data (such as land use types, GDP density, and population density) are obtained from the Data Centre for Resources and Environmental Sciences, Chinese Academy of Sciences (RESDC) (http://www.resdc.cn).

3. Qinghai-Tibet Plateau boundary data are from the study "On the Extent and Area of the Qinghai-Tibet Plateau: Geographic Information System Data on the Boundaries and Area of the Qinghai-Tibet Plateau" published by Zhang et al. (2014) (http://www.geodoi.ac.cn/doi.aspx?doi=10.3974/geodb.2014.01.12.v1).

---

## [Editor Report · Decision Letter 2]

18 Feb 2025

Nonlinear variations and drivers of vegetation NPP on the Tibetan Plateau: interaction of natural and human factors

PONE-D-24-48281R2

Dear Dr. peng,

We’re pleased to inform you that your manuscript has been judged scientifically suitable for publication and will be formally accepted for publication once it meets all outstanding technical requirements.

Kind regards,

Bao Yang, Ph.D, Prof.

Academic Editor

PLOS ONE
---

## [Editor Report · Acceptance letter]

PONE-D-24-48281R2

PLOS ONE

Dear Dr. peng,

I'm pleased to inform you that your manuscript has been deemed suitable for publication in PLOS ONE. Congratulations! Your manuscript is now being handed over to our production team.

Kind regards,

on behalf of

Dr. Bao Yang

Academic Editor

PLOS ONE